# Dimethylsulfide (DMS), marine biogenic aerosols and the ecophysiology of coral reefs

Rebecca L. Jackson[1,2*], Albert J. Gabric[2,3], Roger Cropp[1] and Matthew T. Woodhouse[4]

[1]School of Environment and Science, Griffith University, Gold Coast, QLD, Australia
[2]Australian Rivers Institute, Griffith University, Gold Coast, QLD, Australia
[3]School of Environment and Science, Griffith University, Nathan, QLD, Australia
[4]Climate Science Centre, Oceans and Atmosphere, Commonwealth Scientific and Industrial Research Organisation, Aspendale, VIC, Australia

*Correspondence to:* Rebecca Jackson (rebecca.jackson7@griffithuni.edu.au)

**Abstract.** Global climate change and the impacts of ocean warming, ocean acidification and declining water quality are adversely affecting coral reef ecosystems. This is of great concern as coral reefs provide numerous ecosystem, economic and social services. Corals are also recognized as being amongst the strongest individual sources of natural atmospheric sulfur, through stress-induced emissions of dimethylsulfide (DMS). In the clean marine boundary layer, biogenic sulfates contribute to new aerosol formation and the growth of existing particles, with important implications for the radiative balance over the ocean. Evidence suggests that DMS is not only directly involved in the coral stress response, alleviating oxidative stress, but may create an "ocean thermostat" which suppresses sea surface temperature through changes to aerosol and cloud properties. This review provides a summary of the current major threats facing coral reefs and describes the role of dimethylated sulfur compounds in coral ecophysiology and the potential influence on climate. The role of coral reefs as a source of climatically important compounds is an emerging topic of research, however the window of opportunity to understand the complex biogeophysical processes involved is closing with ongoing degradation of the world's coral reefs. The greatest uncertainty in our estimates of radiative forcing and climate change are derived from natural aerosol sources, such as marine DMS, which constitutes the largest flux of oceanic reduced sulfur to the atmosphere. Given the increasing frequency of coral bleaching events, it is crucial that we gain a better understanding of the role of DMS in local climate of coral reefs.

# 1. Introduction

Tropical corals grow in warm, oligotrophic waters from approximately 30°N to 30°S (Fig. 1). This geographical restriction is due to the physiological requirements of reef-building Scleractinian corals which have relatively narrow thermal, light and salinity tolerance ranges (Bourne et al., 2016; Lesser, 2010; Hughes et al., 2018). The term coral holobiont refers to the symbiosis of the coral host with multiple endosymbiotic microorganisms including cyanobacteria, which aid in nitrogen fixation (Lesser et al., 2004) and photosynthetic dinoflagellates within the Symbiodiniaceae family (LaJeunesse et al., 2018; Muscatine and Porter, 1977). Several genera of Symbiodiniaceae may exist within the coral host depending on species and thermo-tolerance and are collectively termed zooxanthellae. These are acquired via phagocytosis and reside within membrane-enclosed compartments (the symbiosome) within the coral gastrodermis (Tresguerres et al., 2017). Zooxanthellae provide corals with their vibrant colours (e.g. Fig. 2) and 95% of their metabolic requirements via translocation of photosynthetically fixed carbon and in return, depend upon the coral host to obtain inorganic carbon (Falkowski et al., 1984; Dubinsky and Falkowski, 2011). This symbiosis is essential for coral survival (Bourne et al., 2016).

Shallow-water coral reefs cover only 0.1% (~600,000 km$^2$) of the marine environment (Spalding et al., 2001), yet provide numerous essential ecosystem, economic and social services (Barbier et al., 2011; Hoegh-Guldberg et al., 2007; Jones, 2015), with an estimated value of US$9.9 trillion annually (Costanza et al., 2014). Approximately one third of all described marine species obtain food and habitat from coral reefs (Reaka-Kudla, 1997). This biodiversity makes them hotspots for global tourism, providing approximately US$36 billion annually (Spalding et al., 2017) and for the discovery of new medically important biochemical compounds (Kumar, 2006). The high productivity of coral reefs also provides a valuable source of income for global fisheries, estimated at US$5.7 billion per year (Cesar et al., 2003). Fringing and barrier reefs are also highly effective at mitigating coastal erosion by reducing wave energy by up to 97% (Ferrario et al., 2014). Coral reefs provide many other services including nutrient cycling (Barbier et al., 2011; Gattuso et al., 1998; Bourne et al., 2016) and potentially climate regulation through stress-induced emissions of volatile sulfur compounds (Cropp et al., 2018; Fischer and Jones, 2012; Jones, 2015; Fiddes et al., 2018).

When corals experience physiological stress from high sea surface temperature (SST), irradiance, hyposalinity or exposure to air at low tide, they increase production of dimethylsulfoniopropionate (DMSP), which acts as an antioxidant for the coral holobiont (Deschaseaux et al., 2014a; Hopkins et al., 2016; Gardner et al., 2016). Depending on the degree of coral stress, a portion of DMSP is enzymatically cleaved to form the volatile gas dimethylsulfide (DMS) (Raina et al., 2009; Bullock et al., 2017). Upon ventilation to the atmosphere, DMS is oxidised to form aerosol precursors (Andreae and Crutzen, 1997), which can affect aerosol and cloud properties (Sanchez et al., 2018; Charlson et al., 1987). DMS emissions from coral reefs may therefore exert a significant influence on local climate.

The role of coral reefs in climate has only recently begun to be appreciated, despite zooxanthellate corals being amongst the largest individual sources of biogenic sulfur in the marine environment (Burdett et al., 2015; Broadbent and Jones, 2004; Haydon et al., 2018; Swan et al., 2017a; Van Alstyne et al., 2009). However, our understanding of the biogeochemical and ocean-atmosphere interactions involved in the coral reef DMS cycle is not yet complete (Jones, 2015), contributing to the large uncertainty in the role of natural aerosols in our estimates of radiative forcing (Carslaw et al., 2013). The ongoing degradation of the world's coral reefs provides an urgency to gaining a better understanding of these processes. A concern is whether a decline in emissions of DMS-derived aerosol will alter the local radiative balance and accelerate warming in coral reefs, impeding the ability of corals to cope with future climate change. In this review we examine the role of DMS in coral ecophysiology and the emerging topic of coral reefs as a source of marine biogenic aerosol (MBA). The implications of coral bleaching and ongoing coral reef degradation will also be discussed to highlight the importance of a multi-disciplinary approach to coral reef management.

## 2. The role of dimethylated sulfur compounds in coral reefs

### 2.1 The coral reef sulfur cycle

Scleractinian corals are recognized as being a significant source of DMS/P, with reported concentrations of 54,381 nmol DMSP in a 50 mL sample of coral mucous ropes (Broadbent and Jones, 2004), 409-459 nmol DMSP $cm^{-2}$ of coral surface (Frade et al., 2016; Hill et al., 1995, 2010), and 45.9 nmol $m^{-3}$ atmospheric DMS ($DMS_a$) above a coral reef exposed to air at low tide (Swan et al., 2017a). In the coral reef-dense region of the Indo-pacific, *Acropora* spp. are abundant and contain the highest reported concentrations of DMS/P amongst coral genera (Swan et al., 2017b), exceeding 3,500 nmol DMSP $cm^{-2}$ (Broadbent et al., 2002). Other zooxanthellate organisms including octocorals (soft coral) and giant clams contain high amounts of DMSP, with concentrations of up to 4710 nmol $mg^{-1}$ protein (Haydon et al., 2018) and 33,000 nmol $g^{-1}$ tissue (Hill et al., 2000), respectively. Concentrations in the coral holobiont and other zooxanthellate organisms are substantially higher than that reported for benthic macroalgae (1.5 nmol $cm^{-2}$) and individual free-living dinoflagellates (1.5 x $10^{-4}$ nmol zooxanthellae$^{-1}$) (Broadbent et al., 2002).

In the coral holobiont, the biosynthesis of DMSP is upregulated in response to physiological stress. DMSP catabolism occurs via the cleavage and demethylation pathways, yielding DMS or methanethiol (MeSH), respectively (Fig. 3). The cleavage pathway is mediated by either DMSP-lyase enzymes, yielding DMS and acrylate (Bullock et al., 2017; Caruana and Malin, 2014; Raina et al., 2009), or via the addition of an acyl coenzyme A (encoded for by the microbial dddD gene) before cleavage to DMS occurs (Todd et al., 2007). The amount of DMS ultimately released to surrounding reef waters is dependent upon the ratio of DMSP breakdown to DMS and DMS/P photo-oxidation to dimethyl sulfoxide (DMSO) by reactive oxygen produced by the coral holobiont under stress (Fig. 3). Although oxidation to DMSO is a sink of DMS/P, DMSO may also be reduced back to DMS and therefore also acts as a DMS source in coral reef waters (Fischer and Jones, 2012; Gardner et al., 2016;

Deschaseaux et al., 2014b). The physiological requirements of microbial communities within the coral host and in coral reef waters also plays an important role in the cycling of DMSP, switching between the DMSP cleavage and demethylation pathways (Fig. 3). Microbial cleavage of DMSP yields DMS (and acrylate via the DMSP-lyase cleavage pathway), which serves as an important carbon source, while the demethylation pathway yields an organic sulfur source for the coral microbiome (Bourne et al., 2016; Sun et al., 2016; Bullock et al., 2017).

DMSP is a zwitterion and will not diffuse across cell membranes. Corals expel their endosymbiotic microalage at a rate of 0.2-0.4% zooxanthellae cells $day^{-1}$ in response to elevated irradiance or temperature (Jones et al., 2007). DMSP in the form of dissolved ($DMSP_w$) or particulate ($DMSP_p$), is instead released into surrounding waters from expelled zooxanthellae (Fig. 3), via natural cell senescence or grazing by zooplankton and herbivorous fish (Dacey and Wakeham, 1986). Dissolved DMS ($DMS_w$) and dissolved DMSO ($DMSO_w$) may also be released in zooxanthellae exudates or in coral mucous (Broadbent and Jones, 2006; Raina et al., 2009). $DMS_w$ is then ventilated to the atmosphere where it has a residence time of approximately one day (Khan et al., 2016), before atmospheric reaction mechanisms such as oxidation to sulfur dioxide ($SO_2$) or methane sulfonic acid (MSA) occur (Andreae and Crutzen, 1997; Barnes et al., 2006; Veres et al., 2020).

Figure 4 provides a simplified overview of the role of dimethylated sulfur compounds in the DMS-aerosol-cloud feedback over coral reefs. When pre-existing aerosol concentrations and cloud cover are high, for example when high wind speeds enhance sea-spray aerosol (SSA) emission, heterogenous oxidation of DMS-derived sulfates occurs rapidly in cloud droplets, contributing to the growth of existing particles rather than the formation of new particles (Woodhouse et al., 2013; Hoffmann et al., 2016). Conversely, during calm, clear conditions, DMS-derived sulfates may undergo further oxidation to sulfuric acid ($H_2SO_4$), followed by gaseous phase nucleation to form new non-sea salt sulfate (nss-$SO_4$) particles (Fig. 4). Nucleation of $H_2SO_4$ may occur within the marine boundary layer (MBL) or in the free troposphere where conditions are more favourable, with entrainment providing an important source of new sulfate particles to the MBL (Sanchez et al., 2018). These secondary aerosols can be efficient cloud condensation nuclei (CCN) and may affect cloud albedo, lifetime and cover over oceans (Charlson et al., 1987).

## 2.2 Mechanisms of biosynthesis

The biochemical pathways involved in DMSP biosynthesis in corals are complex (Bullock et al., 2017). Until recently, it was thought that biosynthesis in the coral holobiont was limited to photosynthetic endosymbionts. However, recent findings show that coral hosts themselves are substantial sources of DMSP (Raina et al., 2013). In phytoplankton, DMSP is produced via a series of four enzymes (Gage et al., 1997) which assimilate sulfur into cysteine and methionine and subsequently into the stable, soluble form DMSP (Bourne et al., 2016; Stefels, 2000). *Acropora* spp. contain orthologues of genes responsible for the expression of two of these enzymes, which encode NADPH-reductase and Ado-Met-dependent methyltransferase enzymes involved in the second and third steps of the DMSP biosynthesis pathway, respectively (Raina et al., 2013). The

methyltransferase enzyme is particularly specific to this pathway and is highly expressed in *Acropora* juveniles which have not yet assimilated endosymbiotic Symbiodiniaceae. Expression declines with adult development likely due to coral association with zooxanthellae (Raina et al., 2013). Additionally, high levels of intracellular DMSP are reported in *Acropora* spp., with *A. tenuis* and *A. millepora* juveniles exhibiting 65% and 76% increases in DMSP concentration in response to thermal stress (Raina et al., 2013). Similar responses were observed in adult corals after exposure to temperatures of 32°C for 10 days. Despite a decline in zooxanthellae density of 84% (where the remaining 16% were severely structurally compromised and not producing DMSP) DMSP concentration increased by 68%, suggesting that the coral polyp was the source of biosynthesis (Raina et al., 2013).

There are a number of hypotheses as to why corals synthesize DMSP. Sulfur is an essential nutrient for all lifeforms, involved in amino acid and protein synthesis. Carbon is also an essential component of life, providing an energy source for respiration. DMSP provides an abundant organic source of both sulfur and carbon in coral reefs and biosynthesis is thought to play a role in the structuring of the coral microbiome (Bourne et al., 2016; Raina et al., 2010; Raina et al., 2009). These endosymbionts provide a number of services to the coral host, including carbon and nitrogen fixation (Falkowski et al., 1984; Dubinsky and Falkowski, 2011) and disease prevention via production of antimicrobial compounds such as tropodithietic acid (TDA), derived from microbial DMSP catabolism (Raina et al., 2016). DMSP is also involved in alleviating oxidative stress in Scleractinian corals (Hopkins et al., 2016; Gardner et al., 2016; Deschaseaux et al., 2014b).

### 2.3 The coral antioxidant response

Empirical evidence shows that DMSP biosynthesis in Scleractinian corals is upregulated during periods of elevated SST, irradiance, aerial exposure at low tide and hyposalinity associated with rainfall or fluvial discharges from adjacent river systems (Andreae et al., 1983; Swan et al., 2017a; Broadbent and Jones, 2006; Jones et al., 2007; Gardner et al., 2016; Hopkins et al., 2016; Raina et al., 2013). Corals and the endosymbiotic relationship they depend upon, have a relatively narrow thermal, light and salinity tolerance range (Lesser, 2010; Nielsen et al., 2018). Enhanced production of dimethylated sulfur compounds when these tolerance ranges are approached suggests DMSP may play a key role in the coral stress response. Strong production of DMS by corals exposed to air at low tide was first noted by Andreae et al. (1983) during ship measurements throughout the Florida Keys. Although $DMS_w$ production was dominated by benthic and planktonic algae, the coral reef became a much stronger source of DMS during low tide, with spikes in $DMS_a$ more than twice that of the background oceanic signal (Andreae et al., 1983). This was more recently observed in the Great Barrier Reef (GBR), Australia, where $DMS_a$ can exceed 45 nmol m$^{-3}$ during low tide (Swan et al., 2017a). Field measurements show that DMSP and DMS are correlated with SST (Jones et al., 2007) and tide height (Jones and Trevena, 2005) throughout the GBR. This relationship has also been demonstrated in chamber experiments (Raina et al., 2013; Hopkins et al., 2016). For example, *A. intermedia* sampled from Heron Island in the southern GBR increased intracellular DMSP production by 45% in response to a 2°C rise in ambient SST (Jones et al., 2014). Similarly,

in Curaçao reef-building corals the biosynthesis of a number of betaines including DMSP, is modulated by irradiance, indicating a role in photoprotection in the coral holobiont (Hill et al., 2010).

Elevated irradiance and/or SST can impair zooxanthellae photosystems, destabilizing the photosynthetic electron-transport chain and increasing the production of harmful reactive oxygen species (ROS) (Lesser et al., 1990; Jones et al., 2002; Yakovleva et al., 2009; Downs et al., 2002). These ROS can diffuse from the symbiosome into the coral gastrodermis where antioxidant defences may prevent oxidative damage by reducing ROS levels to the tolerance threshold of the coral holobiont. In phytoplankton, superoxide dismutase (SOD) and glutathione interactions form an antioxidant pathway, whereby SOD converts superoxide anions to hydrogen peroxide ($H_2O_2$) and oxygen ($O_2$) while the glutathione pathway regenerates the enzyme ascorbate peroxidase, responsible for scavenging $H_2O_2$ (Lesser, 2006). DMSP, DMS and acrylate are also capable of scavenging ROS (Sunda et al., 2002) and may therefore form a similar antioxidant system in corals (summarised in Fig. 3). Furthermore, Curaçao corals contain high concentrations of other photoprotective betaines, with collective total concentrations ranging from 12-204 (mean 75) nmol $L^{-1}$ of coral tissue (Hill et al., 2010). Glycine betaine and proline betaine were the most prominent compounds in the sampled corals, and may too, increase the oxidative stress threshold of the coral holobiont.

The role of DMSP in the coral antioxidant response is supported by several studies which show that when stress exceeds coral physiological limits (e.g. SST > 30°C, salinity < 24 psu or exposure to air at low tide), DMSP and DMS concentrations decline as the rate of oxidation to DMSO increases (Hopkins et al., 2016; Gardner et al., 2016; Deschaseaux et al., 2014b). For example, when *Acropora* spp. were exposed to SST two degrees above the climatological summer maximum for a period of 36 hours, chamber headspace DMS concentrations declined by 93% as the ratio of dissolved DMSO:DMSP increased (Fischer and Jones, 2012). Similarly, increases in $DMS_a$ occur when corals are exposed to air at low tide, due to direct atmospheric exchange (Swan et al., 2017a). $DMS_a$ declines with time as oxidation to DMSO increases. When corals are resubmerged, a second spike in $DMS_a$ occurs as the dissolution of DMS-rich coral mucous increases sea surface concentration and ventilation to the atmosphere. Atmospheric and dissolved DMS levels then decline again as DMSO concentrations increase (Hopkins et al., 2016; Swan et al., 2016). This trend is also observed when corals are exposed to low salinity (~16 psu) (Gardner et al., 2016), as occurs on inshore continental reefs during the wet season when fluvial discharge is high. The decline in intracellular DMSP and DMS emissions with high stress levels reflects enhanced photochemical oxidation to DMSO, suggesting DMS/P is a sensitive indicator of coral stress and an important antioxidant in the coral holobiont. As mentioned earlier, soft corals (e.g. Octocorallia spp.) contain large quantities of DMS/P. However, unlike Scleractinian corals, emissions of DMS from zooxanthellate soft corals do not vary seasonally, perhaps indicating an alternative ecological role involving holobiont community structuring (Haydon et al., 2018). Given the increasing dominance of soft corals in disturbed coral reef systems (Inoue et al., 2013; Nörstrom et al., 2009), a shift in coral community structure could lead to a significant change in DMS emissions from coral reefs.

## 3. Environmental stressors and their impact on DMSP cycling

A key question is how the DMS/P antioxidant system in corals will respond to a changing climate, and whether a change in DMS/P synthesis facilitates or hinders corals' ability to adapt to rapid environmental change. Bourne et al. (2016) review the cumulative impacts of coral reef stressors, and claim that when temperature, ocean acidification, water quality and other stressors such as overfishing accumulate, the diversity and resilience of the coral microbiome and coral reef ecosystem declines. Reduced diversity may lead to a higher risk of coral disease and mortality and wide-scale ecosystem shifts before a new stable state is achieved (Bourne et al., 2016). Figure 5 summarises the impacts of climate change and modification of land-use on coral reefs and the possible impact on DMS emissions. However, the impacts of ocean warming, ocean acidification and declining water quality on net DMS production and sea-air flux in coral reefs remains uncertain.

### 3.1 Ocean warming

Ocean warming is considered to be one of the greatest threats to coral reefs (Baker et al., 2008; Heron et al., 2016; Hughes et al., 2018; Skirving et al., 2019). More than 90% of the heat energy accumulated in the Earth's climate system, largely a result of anthropogenic greenhouse gas (GHG) emissions, is stored in the ocean. Consequently, global mean SST has risen by 0.5°C over the last 40 years (Intergovernmental Panel on Climate Change (IPCC), 2014). The stability of the coral system is dependent upon the range of temperature and irradiance experienced by the coral host and their endosymbionts (Dubinsky and Falkowski, 2011; Lesser, 2010). If thermal and/or irradiance stress induces the production of excess ROS (Lesser, 2010), corals may change the composition of zooxanthellae clades (symbiont switching or shuffling) in an attempt to reduce oxidative stress (Bay et al., 2016), or expel their zooxanthellae resulting in coral bleaching (Gates et al., 1992; Nielsen et al., 2018; Lesser et al., 1990). Temperature and irradiance stress can be exacerbated during exposure to air at low tide during the day, which can result in extensive partial mortality of intertidal corals (Buckee et al., 2019). The severity of coral bleaching and subsequent mortality is dependent on the magnitude of stress and duration of exposure (Ainsworth et al., 2016).

As many Scleractinian corals now reside in regions with SST close to their upper physiological limits, a 0.5°C rise in SST above the local summer maximum for a period of several weeks is sufficient to cause coral bleaching and mortality (Berkelmans, 2009). SST greater than ~2°C above the summertime maximum can result in coral bleaching over much shorter time-scales, depending on coral tolerance to thermal and/or irradiance stress and on the magnitude and duration of exposure (Bainbridge, 2017). If stress levels subside quickly, corals may regain their zooxanthellae and recover, although surviving corals can have reduced growth, calcification and reproductive rates and a higher incidence of disease and competitive susceptibility (Baker et al., 2008; Ward et al., 2002; Chaves-Fonnegra et al., 2018). Recent work found an 89% decline in larval recruitment during the 2018 spawning event in the GBR after consecutive mass bleaching events adversely affected populations of adult spawning corals (Hughes et al., 2019). If current GHG emissions continue, it is estimated that 50% of

coral reefs will experience annual severe coral bleaching by 2030, and more than 95% by 2050 (Burke et al., 2011), assuming corals cannot adapt or acclimate to the changing climate.

Empirical evidence has demonstrated that DMS/P biosynthesis is upregulated during thermal stress, followed by oxidation to DMSO in temperature-sensitive species (e.g. Deschaseaux et al., 2014a). Therefore, warmer oceans will likely cause an upregulation of DMSP biosynthesis and oxidation to DMSO in coral reefs (Fig. 5). However, variation among coral species and zooxanthellae type and their interactions with marine macro- and microalgae will govern changes to the coral reef sulfur cycle. When corals experience thermal stress, *Acropora* spp. in the GBR have been found to expel thermo-sensitive Clade C Symbiodiniaceae, instead taking up the more tolerant Clade D variety (Bay et al., 2016). Coral switching and/or shuffling of endosymbionts may occur rapidly over time-scales of several days to weeks, or occur gradually over several months in response to changes in environmental conditions. Changing their endosymbiont community structure to favour more tolerant species can assist corals in coping with physiological stress, as thermo-tolerant zooxanthellae photosystems are not producing excess amount of ROS. For example, symbiont shuffling to favour type D symbiont dominance in *A. millepora* increased thermo-tolerance by ~1.5°C (Berkelmans and Van Oppen, 2006). Interestingly, temperature-tolerant zooxanthellae clades (e.g. clade D) produced less DMSP compared to temperature-sensitive clades when exposed to the same conditions (Deschaseaux et al., 2014b; Bay et al., 2016), likely reflecting differences in thermo-tolerance. However, tolerance thresholds within zooxanthellae clades are highly variable (Klueter et al., 2017) and do not always predict DMSP biosynthesis (Steinke et al. 2011).

Marine micro- and macroalgae compete with corals for light, nutrients and space. When coral reefs are degraded by coral bleaching, fleshy macroalgae such as *Polysiphonia* spp. and *Ulva* spp. which also synthesize DMSP (Van Alstyne and Puglisi, 2007), may dominate (De'ath and Fabricius, 2010; Bell, 1992; McClanahan et al., 2011; McCook and Diaz-Pulido, 2002). Phytoplankton blooms, which are additional sources of DMSP in coral reefs, have been positively correlated with aerosol concentration in the Southern Ocean, reflecting an enhanced biogenic particle source from DMS and other organic emissions (Korhonen et al., 2008; McCoy et al., 2015). This additional algal source of biogenic sulfur may counteract any decline in coral emissions (Fig. 5). Algal blooms may also increase light attenuation at the surface, alleviating coral light stress however, these organisms also promote heat absorption at the ocean surface, converting ~60% of absorbed photons to heat (Lin et al., 2016). Therefore, increasingly algal-dominated coral reef communities may exacerbate the warming effects of GHGs in coral reefs.

**3.2 Ocean acidification**

Ocean acidification (OA) affects coral reefs by enhancing dissolution and limiting pH-sensitive coral calcification rates (Tresguerres et al., 2017; Albright et al., 2018; Steiner et al., 2018). Anthropogenic carbon dioxide ($CO_2$) emissions are $10.1 \pm 0.5$ Gt C year$^{-1}$ (Le Quéré et al., 2018), of which approximately 30% is absorbed by the oceans and affect seawater chemistry

(IPCC, 2014; Orr, 2011). Increased $CO_2$ levels enhance the formation of carbonic acid ($H_2CO_3$) which readily dissociates into bicarbonate ions ($HCO_3^-$) and protons ($H^+$) as per Eq. (1). These protons react with carbonate ($CO_3^{2-}$) to produce more $HCO_3^-$ ions, which in turn, decrease the bioavailability of $CO_3^{2-}$ for marine calcification (Orr, 2011). The current concentration of $CO_3^{2-}$ ions in the ocean is 200 µmol kg$^{-1}$ of seawater (Zeebe and Tyrrell, 2019), around 40 µmol kg$^{-1}$ less than the minimum over the past 420,000 years (Hoegh-Guldberg et al., 2007). OA also favours erosion of calcareous structures (Eq. (2)), with massive *Porites* corals in the northern GBR displaying annual declines in linear extension rates and skeletal density of 1.02% and 0.36% respectively (Cooper et al., 2008). If current GHG emissions persist, it is estimated that less than 15% of the world's tropical coral reefs will be in regions with carbonate saturation sufficient for coral growth by 2050 (Burke et al., 2011).

$$CO_{2\,(g)} + H_2O_{\,(l)} \Leftrightarrow H_2CO_{3\,(aq)} \Leftrightarrow HCO^-_{3\,(aq)} + H^+_{(aq)} \Leftrightarrow CO^{2-}_{3\,(aq)} + 2H^+_{(aq)} \tag{1}$$

$$CaCO_{3\,(s)} + 2H^+_{(aq)} \Leftrightarrow Ca^{2+}_{(aq)} + 2HCO^-_{3\,(aq)} \tag{2}$$

It is unclear how DMSP cycling will be influenced by OA and this is particularly true of coral reefs. In remote oceans such as the Arctic, biosynthesis of DMSP is predicted to increase due to enhanced phytoplankton biomass and availability of inorganic carbon (Archer et al., 2013). Other studies predict that the flux of DMS to the atmosphere will decrease due to a decline in the abundance of DMSP-producers such as phytoplankton (Schwinger et al., 2017; Archer et al., 2018). Regardless, acidification appears to be adversely affecting coral calcification and health (Cooper et al., 2008; Hoegh-Guldberg et al., 2007; Orr, 2011; Albright et al., 2018). If this continues, a decline in coral cover may result in a decline in DMS emissions from coral reefs (Fig. 5). Again, opportunistic growth of marine algae is predicted to dominate degraded coral reef systems (De'ath and Fabricius, 2010; Brodie et al., 2011; McCook and Diaz-Pulido, 2002). It is possible that increased DMS emissions from this algal source may counteract the decline in coral emissions (Fig. 5). However, recent evidence from the southern GBR demonstrated that the combined effects of ocean warming and acidification under RCP8.5 conditions (an IPCC representative concentration pathway (RCP) whereby GHG and aerosol emissions drive an 8.5 W m$^{-2}$ increase in radiative forcing by 2100) also impaired calcification in the macroalga *Halimeda heteromorpha* (Brown et al., 2019). Other studies have demonstrated that temperature has a stronger influence on DMS/P production in cultured microalgae, whereby increased production in response to temperature outweighed the decline in DMS/P biosynthesis due to OA (Arnold et al., 2013).

### 3.3 Water quality and eutrophication

Declining water quality is another cause of coral reef degradation. Although inner-shelf coral reefs exposed to reduced water quality are more resistant to coral bleaching, resilience is low due to increased susceptibility to disease and predation (MacNeil et al., 2019). Eutrophication arises when enhanced loading of the limiting macro-nutrients nitrogen (N) and/or phosphorous (P) induce excessive growth of marine algae (Howarth et al., 2011; Diaz and Rosenberg, 2008; McEwan et al., 1998), which

impede coral growth and reduce the opportunity for new corals to establish (De'ath and Fabricius, 2010; Bell, 1992; McClanahan et al., 2011). This is often caused by catchment runoff and soil erosion from land clearing, agriculture and urbanization (Brodie et al., 2011; McKergow et al., 2005). Resulting algal blooms can reduce water clarity, deplete dissolved oxygen and release toxins (Osborne et al., 2001), all of which adversely affect coral health and grazing fish populations.

Depletion of grazing fish populations further favours algal growth by removing top-down control (McClanahan et al., 2011). Eutrophication leading to an increase in phytoplankton standing stock also contributes to outbreaks of *Acanthaster planci* (Crown of Thorns Seastar), an invasive predator of corals and a significant threat to coral reefs (Fabricius et al., 2010; De'ath et al., 2012; Wooldridge and Brodie, 2015). These impacts are compounded by enhanced sedimentation from riverine discharge and dredging, which cause benthic smothering, increased turbidity and reduced light penetration, all of which

adversely affect photosynthesising organisms including coral zooxanthellae (Erftemeijer et al., 2012). Sea-level rise and changes to large-scale oceanic circulation in response to global warming is exacerbating coastal erosion, which is contributing to declining water quality. Climate change has also caused an increase in storm surges which further degrade coral reefs (Mellin et al., 2019). Measurements of dissolved DMS/P have been found to decline along a gradient of relatively pristine to human-impacted coral reefs in the central GBR (Jones et al., 2007). However, it is unclear whether poor water quality will result in a

net decline in DMS flux from coral reefs in future, as an enhanced algal source may counteract declining emissions from corals (Fig. 5).

## 4. Marine biogenic aerosols

### 4.1 The CLAW hypothesis

The CLAW hypothesis (1987) proposed that marine DMS emissions increase the formation of low-level, high albedo cloud

cover, establishing a biologically derived negative feedback on the warming effects of GHG (Charlson et al., 1987; Shaw, 1983). Despite decades of research, our knowledge of the biological, chemical and atmospheric processes involved in the biosynthesis, flux and climatic influences of DMS remains incomplete. Some suggest that the original hypothesis is an over-simplification (Green et al., 2014) or no longer relevant (Quinn and Bates, 2011) with anthropogenic perturbation of the atmosphere throughout much of the globe (Spracklen and Rap, 2013). However, others remain steadfastly positive about the

role of DMS in global climate (Grandey and Wang, 2015).

### 4.2 DMS sea-to-air flux

The global sea-air flux of DMS is estimated to be 17.6 - 34.4 Tg S year[-1] (Kettle and Andreae, 2000; Lana et al., 2011; Land et al., 2014), accounting for approximately 50% of the natural global atmospheric sulfate burden (Bates et al., 1992; Simó, 2001; Barnes et al., 2006). The large DMS flux range reflects the uncertainty in undersampled ocean regions where newly

collected data is improving the estimate (Webb et al., 2019). Emissions are highly variable in both space and time and are primarily governed by ocean biology in the mid-low latitude oceans (Broadbent and Jones, 2006; Jones et al., 2007; Korhonen

et al., 2008) and sea-ice dynamics in polar regions (Gabric et al., 2018). The sea-to-air flux of DMS depends on its surface ocean concentration (Lana et al., 2011), SST (Yang et al., 2011), wind speed (Ho and Wanninkhof, 2016) and water depth or tide height (Swan et al., 2017a). Higher sea surface DMS concentration, SST and wind speed (which increase diffusivity), and reduced tide height, will enhance the sea-air flux of DMS in coral reef waters.

It is unclear how marine DMS production and sea-air flux will change in response to climate change. In the Arctic, DMS flux is estimated to increase by 86% to 300% under modelled scenarios of three to four-times present-day $CO_2$ concentrations. This predicted increase is due to a combination of enhanced biological activity and changes to sea-ice dynamics (Gabric et al., 2005; Qu et al., 2017). On a global scale, atmospheric DMS concentrations are predicted to increase by 41% with a tripling of

10 atmospheric $CO_2$, increasing mean aerosol optical depth (AOD) by 3.5% and cooling the northern and southern hemisphere by 0.4 K and 0.8 K respectively (Gabric et al., 2013). Other studies simulate a global reduction of the DMS flux by 10-18% (Kloster et al., 2017; Six et al., 2013) and 48% (Schwinger et al., 2017) under various ocean acidification scenarios by the end of the 21st and 22nd centuries, respectively. Modelled scenarios predict an additional global warming of 0.23-0.48 K due to a decline in DMS-derived sulfate aerosols and cloud albedo (Schwinger et al., 2017; Six et al., 2013), although significant

regional variability in the response of DMS emissions to climate perturbations was present. In the Southern Ocean, DMS emissions increased in response to warming, counteracting the predicted regional decrease in emissions in response to OA (Schwinger et al., 2017). This regional variability is likely driven by differences in carbonate chemistry and conditions to which phytoplankton communities are adapted to, which play an important role in phytoplankton community structure and DMS production (Hopkins et al., 2018). Overall, studies from the last few decades of global biogeochemical modeling and

mesocosm experiments provide both negative and positive changes in DMS flux under future climate conditions and highlight the need for an improved understanding of regional biogeochemical responses to climate change. This is particularly true of coral reefs, where few studies have attempted to quantify DMS emissions under current or future climate conditions.

The total contribution of coral reefs to the global sulfur budget is not yet clear, however, the GBR and lagoon waters (347,000

$km^2$) are estimated to emit 0.02 Tg S year$^{-1}$ (Jones et al., 2018). It is noted that coral cover in the GBR is ~20,000 $km^2$ and consequently, DMS flux from the surrounding lagoon would appear to be higher than estimates from the coral reef. Sources of DMS in lagoon waters include phytoplankton, macroalgae and dissolved DMS released from the coral holobiont and other zooxanthellate organisms. Therefore, the total DMS flux can be considered an estimate of coral reef DMS emissions. This is the first estimate of DMS flux from the GBR, however it is based on a limited number of field observations and does not

account for occasional large pulses of DMS emitted by corals during aerial exposure at low tide (Swan et al., 2017a). When these events are considered, DMS sea-air flux from coral reefs would periodically increase. Hopkins et al. (2016) provided an estimate of DMS flux per unit area of the GBR, based on laboratory measurements taken from *A. horrida* periodically exposed to air. As *Acropora* are the dominant genus throughout the Indo-Pacific and are amongst the strongest producers of DMS/P, it was estimated that corals in the GBR release 3 - 11 mmol S m$^{-2}$ day$^{-1}$. According to this estimate the GBR releases 0.01 - 0.04

Tg S yr$^{-1}$, consistent with the estimate provided by Jones et al. (2018). Assuming that DMS flux is constant across coral reefs and lagoon waters, tropical coral reefs (~600,000 km$^2$) emit 0.02-0.08 Tg S yr$^{-1}$. Although this represents < 1% of global or tropical average DMS sea-air flux estimates (Lana et al., 2011), it is a substantial amount of sulfur released from only 0.2% of the ocean surface, with the potential to influence local climate in coral reefs.

## 4.3 Complexity of the climate response to DMS

Once ventilated to the atmosphere, the influence of DMS on climate is dependent on the efficiency of DMS oxidation to $SO_2$ and secondarily to $H_2SO_4$ (Barnes et al., 2006), which ranges from 0.14 to 0.95 in the MBL due to spatiotemporal variation in pre-existing atmospheric properties (Glasow and Crutzen, 2004). The annual mean contribution of DMS to CCN is estimated to be only 3.3% in the northern hemisphere and 9.9% in the southern hemisphere (Woodhouse et al., 2010). Consequently, the sensitivity of CCN to DMS emissions at a global scale is low, with a 1% increase in global DMS flux resulting in only a 0.1% increase in mean CCN (Woodhouse et al., 2010). Although the response of CCN to DMS is low on a global scale, spatial and seasonal estimates vary, with higher responses evident in pristine regions of high biological activity (Woodhouse et al., 2013; Lana et al., 2012; Vallina et al., 2007; Zavarsky et al., 2018). However, an alternate DMS oxidation pathway, involving the formation of hydroperoxymethyl thioformate (HPMTF), was recently observed in the marine atmosphere and accounts for ~30% of oceanic DMS oxidation, with important implications for the representation of DMS-derived sulfates in atmospheric models (Veres et al., 2020). Further, models may be underestimating DMS emissions in some parts of the ocean due to sparsity in global observations, resulting in large uncertainties in interpolated flux estimates (Mulcahy et al., 2018).

In the remote MBL, biogenic aerosols such as DMS-derived sulfates may be important in smaller-scale climate regulation. Vallina et al. (2007) identified a strong sensitivity of fine-mode AOD and CCN number to the rate of DMS$_a$ oxidation in pristine, high latitude regions. In the North Atlantic, new sulfate particles formed via the nucleation of DMS-derived $H_2SO_4$ in the free troposphere accounted for 33% of CCN at 0.1% supersaturation, largely due to enhanced phytoplankton biomass in spring (Sanchez et al., 2018). Similarly, in the Southern Ocean, seasonal variability in cloud droplet number is primarily driven by changes in sea-spray, organic matter and biogenic sulfates, with seasonal maxima during spring and summer, coinciding with enhanced biological activity and DMS emissions. DMS-derived nss-SO$_4$ particles account for 7-20% of mean CCN in winter and 43-65% in summer (Korhonen et al., 2008), increasing the Southern Ocean summertime mean reflectivity by more than 10 W m$^{-2}$ (McCoy et al., 2015). Conversely, complete removal of DMS resulted in a modelled top of atmosphere radiative forcing of +1.7 W m$^{-2}$ (Fiddes et al., 2018).

## 5. Do coral reefs affect the climate?

The GBR provides a valuable study location for the analysis of biogenic aerosols. Given its large spatial extent (2,300 km in length and 60-250 km in width) and southern hemisphere location, the atmosphere over the GBR is considered to be relatively unpolluted. The region also experiences predominant east to south-easterly trade winds year-round, which carry clean, marine

air to the GBR and advect biogenic compounds northward over downwind coral reefs. Consequently, the majority of research examining the role of coral reefs in MBA production has focused on the GBR region.

## 5.1 Influence on particle formation and growth

The ability of corals to influence local atmospheric properties has only recently begun to be appreciated. Bigg and Turvey (1978) measured total atmospheric particle concentration along the north-east Queensland (QLD) coastline from 1974 to 1977 and found that concentrations were up to seven-times higher in maritime air directly over the GBR compared to the seaward side. Three decades later, Leck and Bigg (2008) observed fluctuations in particle concentration at Lizard Island in the northern GBR during September 1998 and July 2005. Nucleation events forming particles with a diameter greater than $> 0.01$ μm, were observed on all study days with concentrations reaching 40,000 $cm^{-3}$ during the warmer September period and up to 4,300 $cm^{-3}$ in July (Leck and Bigg, 2008). There is strong seasonality in both DMS (Jones et al., 2018; Swan et al., 2017a) and AOD in the GBR (Cropp et al., 2018; Jackson et al., 2018). The large seasonal difference in aerosol loading implies a temperature or irradiance-dependent biological influence. Furthermore, Modini et al. (2009) observed nucleation events at Agnes Water in the southern GBR, in which corals were concluded to be the most likely source of precursor compounds. These events occurred in clean, easterly air masses originating in the MBL, when solar irradiance at the sea surface was high (~1,000 W $m^{-2}$). Particles consisted of 40% organics and 60% sulfate. In the strongest nucleation event, Aitken mode ($< 0.1$ μm in diameter) concentration was measured at 3,200 $cm^{-3}$. This event occurred during a NNE wind, accumulating particles of marine origin from upwind coral reefs. The authors concluded that Aitken mode concentration was too low to have been derived from coagulation alone, thus new particles were largely derived from the condensation of low volatility vapours such as DMS-derived $H_2SO_4$ (Modini et al., 2009). Swan et al. (2016) also measured high particle concentration over Heron Island in the southern GBR, which coincided with a peak in DMS emissions from the coral reef flat during calm conditions after a low tide, again suggesting that the coral reef was a source of MBA precursor compounds.

The ability of corals to influence the atmosphere above them has also been investigated using remote-sensing approaches. Lana et al. (2012) found that sulfate aerosols positively correlated with CCN and negatively with cloud droplet radius throughout the mid to high latitude oceans and in some tropical latitudes, where a high density of coral reefs exists. Observations of satellite-derived fine-mode aerosol over parts of the GBR show concentrations to be positively correlated with conditions that are empirically shown to cause a stress response in corals and thus, enhance DMS emissions (Cropp et al., 2018; Jackson et al., 2018). Correlation strength between aerosol and coral light stress (a function of PAR, tide height and water clarity) improved with decreasing wind speed at Heron Island, suggestive of a substantial local MBA source over the southern GBR (Cropp et al., 2018).

Figure 6 shows the seasonal zonal-averaged variation in daily mean (2001 - 2017) total AOD (869 nm) over the GBR, highlighting the seasonality in aerosol loading. This data was acquired at 4 km resolution from the Moderate Resolution

Imaging Spectroradiometer (MODIS) on board the Terra and Aqua satellites (https://oceancolor.gsfc.nasa.gov/). Observations over the Australian continent were excluded, thus Fig. 6b shows only zonal-averaged AOD for the GBR and adjacent Coral Sea (Fig 6a). In the northern half of the GBR, AOD is highest during the winter dry season (May - October). Conversely, the southern half of the GBR displays a marked increase in AOD during the warmer months (November - April) and a decrease in winter (Fig. 6b). It is not clear what is driving this latitudinal variability, however emissions of DMS and other volatile organic compounds (VOCs) such as isoprene, cannot be excluded as a potential driver of seasonal variability. Existing research has linked seasonal variability in aerosol and CCN formation to enhanced biological activity in other remote ocean regions (Gabric et al., 2018; Korhonen et al., 2008). The increase in AOD during summer in the southern GBR is also suggestive of a temperature or light-dependent biogenic influence, potentially from the high biomass of coral reefs below.

**5.2 Influence on low-level clouds, sea surface temperature and surface solar radiation**

The Western Pacific Warm Pool (WPWP) is a coral reef dense region located to the north-east of Australia, where SST reaches an upper limit of ~30°C due to regular pulses of low-level cloud (LLC) cover which closely follow the tidal cycle (Kleypas et al., 2008; Takahashi et al., 2010; Ramanathan and Collins, 1991). It has been posited that coral reef DMS emissions contribute to the formation of this "ocean thermostat", acting to suppress ocean temperatures below coral thermal tolerance thresholds, resulting in few coral bleaching events over the past 25 years (Kleypas et al., 2008; Takahashi et al., 2010). Similarly, although an overall warming trend is apparent along the NE coast of Australia, SST in the far northern GBR is warming at a slower rate compared to southern regions (Lough, 2008; Heron et al., 2016). As in the WPWP, this is potentially due to enhanced LLC which is estimated to account for approximately 30% of the variance in SST over the GBR (Leahy et al., 2013). Figure 7 shows LLC closely aligned over the Swains Reefs in the southern GBR, captured by MODIS Terra on 31st May 2018. Back trajectories computed by the National Oceanic and Atmospheric Administration (NOAA) Air Resources Laboratory HYSPLIT Model show dominant south-westerly winds on this day (Stein et al., 2015; Rolph et al., 2017), potentially explaining the north-eastward migration of LLC in the true colour image (Fig. 7).

Reduced SST and irradiance due to enhanced LLC have also been observed to mitigate mass coral bleaching events. For example, high SST in the summer of 1998 resulted in widespread coral bleaching throughout the tropics (Wilkinson et al., 1999). This event caused severe bleaching and subsequent mortality of 99% of *Pocillopora* corals in the Tuamotu Archipelago, French Polynesia (Mumby et al., 2001a). However, high LLC cover significantly reduced the amount of surface solar radiation at coral reefs in the nearby Society Islands where bleaching did not occur (Mumby et al., 2001b). Similarly, high SST (> 31°C) in 1994 caused coral bleaching at Nelly Bay (Magnetic Island) in the central GBR (Jones et al., 1997) however, no bleaching was observed ~60 km away at Pioneer Bay (Orpheus Island) (Jones et al., 2007). Jones et al. (2017) examined DMS$_a$ before, during and after this bleaching event at both locations. At the unbleached reef, regular pulses of LLC often coincided with pulses of DMS (14-20 $\mu$mol m$^{-2}$ day$^{-1}$). Conversely, at the bleached reef LLC cover and DMS emissions were low when SST exceeded 30°C (Jones et al., 2007) and coincided with high solar irradiance (Jones et al., 2017). Together these findings suggest

that stress-induced emissions of DMS from coral reefs may influence local LLC cover to mitigate stress and prevent coral bleaching. However, spatial variation in this phenomenon is apparent, possibly due to prevailing meteorological conditions or other factors affecting coral reef health as discussed in section 2.

### 5.3 Influence on precipitation

Coral reef-derived aerosol emissions have also been proposed to affect precipitation patterns. Jones (2015) discusses high rainfall in regions lying in the path of SE trade winds travelling over the GBR, implying that emissions of DMS and other organic compounds from the coral reef contributed to cloud droplet formation. Conversely, Fiddes et al. (2018) found that DMS-derived sulfate particles enhanced LLC cover and lifetime and consequently suppressed rainfall in parts of the tropics. When high concentrations of fine particles rapidly grow to CCN activation sizes, cloud droplet size decreases, cloud lifetime increases, and rainfall is suppressed (Rosenfeld et al., 2007; Dave et al., 2019; Singh et al., 2018). However, when CCN minimum size is not met, water vapour remains in the atmosphere, suppressing local rainfall yet enhancing rainfall downwind when conditions for particle growth are more favourable (Andreae and Rosenfeld, 2008; Fan et al., 2018; Grandey and Wang, 2015; Lin et al., 2018; Guo et al., 2016). Changes to the source strength of ultra-fine and fine aerosols may therefore affect climate by either increasing or suppressing rainfall. The possibility of reduced precipitation is a critical question for rainfall-sensitive agriculture in NE Australia with ongoing degradation of the GBR and highlights the need for an improved understanding of the relationship between MBA emissions and their impact on cloud micro-physics.

### 5.4 Other biogenic volatile organic compounds

Although DMSP and its derivatives are the most abundant compounds in the coral reef sulfur cycle (~95% of sulfur compounds), other VOCs such as isoprene, dimethyl disulfide (DMDS), carbon disulfide ($CS_2$) and the recently discovered DMSP variant, dimethylsulfoxonium propionate (DMSOP) are also produced in coral reefs (Swan et al., 2016; Thume et al., 2018). Isoprene is the most abundant biogenic VOC in the atmosphere, largely derived from terrestrial vegetation however, oceanic emissions are also reported from marine algae including Symbiodiniaceae (Exton et al., 2013). Swan et al. (2016) recorded large quantities of isoprene in *A. aspera* mucous, likely produced by expelled coral zooxanthellae. Isoprene may be oxidised in the atmosphere or condenses onto existing particles, thus also contributing to aerosol formation over coral reefs with the potential to affect local climate (Fan and Zhang, 2004; Kroll et al., 2006; Surratt et al., 2007). Photochemistry at the ocean surface is a sink of organic compounds, yet interfacial photochemical reactions also produce an important abiotic source of VOCs, estimated to be 23.2 - 91.9 Tg C yr[-1], contributing approximately 60% of organic aerosol mass in the remote MBL (Brüggemann et al., 2018).

### 6. Outlook and the implications for coral bleaching

The physiological response of corals to increasing light and/or temperature stress is nonlinear. As these environmental factors increase beyond optimal levels, DMS emissions initially increase, but when thermal and/or light stress becomes too great,

DMS emissions from corals essentially shut-down (Jones et al., 2007; Fischer and Jones, 2012). This means that there are less aerosol precursor compounds and potentially fewer secondary aerosols forming over the coral reef. Recent work examined anomalies in fine-mode AOD over the GBR during four mass coral bleaching events which occurred primarily due to marine heatwaves (Jackson et al., 2018). Prior to each bleaching event, when SST was rising and corals were likely emitting more DMS, above average AOD was observed. However, just prior to and during the bleaching events, when corals were likely experiencing physiological stress, AOD declined to normal background levels, or below average levels where the coral reef was severely affected. Although these covarying events may be a coincidence, the synchronous decline in AOD with the onset of coral bleaching suggests a link between coral health and atmospheric properties in the GBR.

This raises some important and concerning questions. Will the nonlinear response of DMS emissions from coral reduce their ability to cope with future temperature rises? If coral reefs significantly affect local atmospheric conditions, what will the consequences of ongoing coral reef degradation and coral bleaching be for local or regional climate? A decline in aerosol and LLC formation over relatively pristine coral reefs such as the GBR could occur, potentially increasing SST and establishing a positive feedback on coral stress.

**7. Future research**

Coral reefs are a significant source of dimethylated sulfur compounds however, there remains substantial uncertainty surrounding the importance of DMS in coral homeostasis and local climate, and what implications ongoing coral reef degradation may have on these complex biogeophysical processes. It is important that future research monitors changes to coral health and community structure, the processes driving DMS biosynthesis and emissions, and characterises the properties of marine aerosols and LLC over coral reefs. Understanding these complex ecological and biogeophysical processes will require a multidisciplinary approach, utilizing field, laboratory and remotely sensed data in conjunction with earth system models.

A key area for future research is the investigation of DMS/P biosynthesis amongst different coral reef taxa. The relative abundance of a range of VOCs has been investigated in some species which occupy coral reefs including *Acropora* spp., *Stylophora pistillata, Favites sp., Lobophytum sp.* (soft coral) and cultured samples of benthic macroalgae and crustose coralline algae (Broadbent et al., 2002; Swan et al., 2016, 2017b). Further studies quantifying the biosynthesis of DMS/P/O in other coral genera, and other DMS-producing lifeforms is needed. These studies should particularly focus on species which currently or are predicted to dominate disturbed coral reef systems after an ecological regime shift, such as temperature-tolerant Scleractinian corals, soft corals (e.g. octocorals) and macroalgae. Further laboratory or mesocosm experiments must also investigate how DMS/P/O biosynthesis varies under multiple, synergistic stressors (e.g. Arnold et al., 2013) and whether a change in biosynthesis will assist corals in coping with future disturbances. For example, it is not

known whether DMS emissions from thermo-tolerant coral species will increase with further ocean warming, or whether ocean acidification and deteriorating water quality will impede coral resilience and result in a decline in emissions.

Long-term, high frequency measurements of seawater and atmospheric DMS concentrations in coral reefs is also needed. Short-term field campaigns measuring $DMS_w$ and $DMS_a$ have been conducted over periods ranging from several days of consistent measurements to several months of sporadic measurements (e.g. Andreae et al., 1983; Jones et al., 2007; Jones et al., 2018; Swan et al., 2017a). These campaigns have been crucial in providing high frequency, detailed data necessary to begin to understand the processes driving DMS emissions from coral reefs. For example, Swan et al. (2017a) identified spikes in $DMS_a$ emitted by corals at low tide at Heron Island during consistent $DMS_a$ sampling over a two-week period. Such periodic emissions may not have been captured without consistent measurements (every ~15 minutes) of local tide height and $DMS_a$ concentrations. This finding supported early indications that corals can be a much stronger, albeit intermittent source of DMS during low tide, compared to the background oceanic signal which is likely dominated by algal emissions (Andreae et al., 1983). However, field surveys are very cost- and resource intensive (reviewed in Hedley et al., 2016). The establishment of an autonomous system (e.g. Dacey, 2010) would alleviate some of these caveats and greatly benefit this field of research. This data would allow diel and seasonal changes in DMS emissions from coral reefs to be investigated more thoroughly and allow long-term changes in emissions strength with ongoing climate change to be determined. Although, field surveys alone cannot always capture the larger scale processes involved in DMS oxidation, particle formation and growth. DMS has at atmospheric residence time of up to 1 day (Khan et al., 2016) and may therefore accumulate and affect aerosol and cloud properties over coral reefs downwind of the emissions source where field sampling may not be occurring. Therefore, it can be difficult to deduce empirical relationships between local DMS emissions and atmospheric properties from field surveys alone.

Remote-sensing approaches are useful in this regard, as they enable rapid, cost and time efficient analysis of a wide range of variables at large temporal and spatial scales and can be directly compared with field observations. For example, long-term, high-frequency observations of fine-mode AOD are available from the Aerosol Robotic Network (AERONET) at Lucinda in the central GBR. Comparison of this dataset with long-term measurements of meteorological conditions and $DMS_a$ above the adjacent coral reef would provide valuable insight into the role of DMS on local atmospheric properties. Furthermore, if empirical relationships between coral physiological stress and DMS/P biosynthesis can be deduced from field and laboratory experiments, a remotely sensed proxy for sea surface DMS concentrations and sea-air flux from coral reefs can be calculated (e.g. Cropp et al., 2018; Jones et al., 2007; Jones et al., 2018), similar to that produced for phytoplankton (Galí et al., 2018). Such proxies could be used in conjunction with long-term data on coral cover and health (e.g. Australian Institute of Marine Science Long-Term Monitoring Program) to estimate the source strength of DMS emissions over large regions such as the GBR under future climate scenarios.

Several remote sensing approaches are currently used by coral reef scientists and managers to monitor environmental conditions in coral reefs. The majority of these focus on SST and PAR anomalies as these are largely considered to be the major stressors to corals and are often the dominant factors contributing to coral bleaching (Jones et al., 2002; Lesser, 2010). The National Oceanic and Atmospheric Administration (NOAA) Coral Reef Watch utilises satellite-derived SST and PAR to produce several measures of coral thermal and light stress, including the Hotspot, Degree Heating Week (DHW) and Light Stress Damage (LSD) products (Liu et al., 2006; Skirving et al., 2017). For example, the DHW product is based on spatially dependent SST anomalies above the maximum climatological summertime SST. Empirical evidence has demonstrated that these remotely sensed products reliably predict the severity and extent of mass coral bleaching (Hughes et al., 2018; Bainbridge, 2017; Berkelmans, 2009; Liu et al., 2003; Skirving et al., 2017, 2019). These are also conditions which are known to affect coral DMS/P biosynthesis. Investigating spatiotemporal trends in conditions that influence DMS/P biosynthesis (e.g. thermal stress, measured as DHW), and atmospheric properties such as AOD, could provide insight into the role of DMS in local climate of coral reefs (e.g. Cropp et al., 2018; Jackson et al., 2018).

This information could then be used to inform regional earth system models, to quantify the role of $DMS_a$ in the formation of sulfate aerosols, CCN and the properties of low-level marine clouds over coral reefs. Currently, DMS emissions from coral reefs are poorly constrained and are not explicitly included in any global DMS climatology. Some modelled studies show weak, or no relationship between $DMS_a$ and atmospheric properties in the tropics, whereas strong relationships have been demonstrated in other ocean regions (e.g. Vallina et al., 2007). This may be due to low sensitivity of aerosol and CCN to DMS in the low-latitudes, or a consequence of poor representation of coral reef DMS emissions in global climatologies. Improving the representation of coral reefs in DMS databases, which are subsequently incorporated into earth system models, would provide insight into the sensitivity of local climate to coral reef DMS emissions. Quantifying the radiative cooling effect of marine biogenic aerosols over coral reefs is increasingly important with ongoing ocean warming and coral bleaching.

## 8. Mitigation strategies

The predicted increase in the frequency and severity of mass coral bleaching events may require biological and/or physical interventional management strategies to conserve coral reefs. One possible method is the propagation of temperature-tolerant coral species throughout coral reef ecosystems (Van Oppen et al., 2015). This may help degraded coral reefs recover from disturbances such as coral bleaching and help to prevent further mass coral bleaching events. As discussed above, our understanding of DMS/P biosynthesis amongst coral taxa is relatively limited and temperature tolerance does not always predict the rate of DMS emissions (Steinke et al., 2011). Although, the propagation of these temperature-tolerant species may counteract the predicted decline in DMS emissions in coral reefs dominated by sensitive coral species with ongoing coral reef degradation.

Other strategies are aimed at mitigating the warming effects of GHGs via physical means. Solar Radiation Management (SRM) strategies essentially mimic natural marine aerosol emissions and involve injecting sea-spray or sulfate particles into the atmosphere over coral reefs to increase the albedo of low-lying marine clouds (Crabbe, 2009; Irvine et al., 2017). In a modelled scenario, injecting 5 Tg $SO_2$ annually into the stratosphere above Caribbean coral reefs reduced SST, irradiance and sea-level rise and resulted in a substantial decline in the number of mass coral bleaching events predicted to occur over the next 50 years (Zhang et al., 2017). Similarly, Kwiatkowski et al. (2015) reported that enhancing $SO_2$ concentrations in the tropical stratosphere reduced SST and the risk of coral bleaching over the next 30 years under an RCP4.5 scenario (where GHG and aerosol emissions drive a 4.5 W $m^{-2}$ increase in radiative forcing by 2100) . Other studies have examined the effect of releasing sea-spray from ocean-based vessels to the MBL over coral reefs to brighten low-level marine clouds. Latham et al. (2013) found that an enhanced source of sea-spray aerosol over the GBR, Caribbean and French Polynesia offset the warming effects associated with a doubling of atmospheric $CO_2$, and this reduced the number of coral bleaching events predicted to occur until the end of this century. An additional benefit of these SRM strategies is the potential moderation of climate hazards (Irvine et al., 2019), such as the reduction in the severity of tropical cyclones with a decline in SST (Zhang et al., 2017; Latham et al., 2012).

Another approach involves the implementation of a biodegradable calcium carbonate surface film over coral reefs. Early trials conducted as part of the Surface Films to Attenuate Light into the Great Barrier Reef project have demonstrated that the film increased light attenuation by 30% and had no adverse effects on coral physiology (Great Barrier Reef Foundation, 2018). An initiative proposed by the Reef and Rainforest Research Centre and QLD tourism industry is investigating the feasibility of pumping deep, cold water from 10 - 30 m over coral reefs with high economic and environmental value. Early pilot studies are currently being conducted to test the feasibility of the technology at Moore Reef off the coast of Cairns (Reef and Rainforest Research Centre, 2017).

Solar geoengineering is a realistic approach which may provide short-term protection for high-value or vulnerable coral reefs from rising temperatures. Some of these approaches mimic marine biogenic aerosol emissions and may counteract a decline in DMS-derived sulfate emissions with ongoing coral reef degradation. However, these approaches are highly cost- and resource intensive, particularly for large coral reef systems such as the GBR and success would depend upon the willingness of governments and/or organisations to continue funding and implementing the technology. These strategies also do not address other coral reef stressors such as ocean acidification and the long-term implications are not yet completely understood (Crabbe, 2009; Irvine et al., 2017). Therefore, there is enormous incentive to improve our understanding of the natural drivers of coral resilience, including the role of DMS/P in alleviating oxidative stress and influencing the radiative balance over coral reefs. Future research must decipher the role of DMS and other VOC emissions in marine aerosol and cloud formation. If DMS significantly influences aerosol and cloud properties and alleviates coral physiological stress, it is vital that we understand how these processes may be affected by ongoing anthropogenic and climate change-related disturbances. Further

research is also needed to understand the long-term implications of solar geoengineering strategies, which may need to be implemented as a last resort to conserve coral reefs.

## 9. Conclusions

There is substantial evidence that coral reefs are strong sources of dimethylated sulfur compounds. These play an important role in alleviating oxidative stress in the coral holobiont resulting from high temperatures, irradiance, aerial exposure and hyposalinity. It is possible that a side-effect of this stress response provides a source of precursor compounds for the formation of secondary sulfate aerosols, with important implications for the radiative balance over coral reefs. There is strong seasonality in both DMS emissions and aerosol loading over the 2,300 km stretch of relatively pristine coral reefs in the GBR - a relationship that is indicative of a substantial biogenic influence on AOD. Field studies have observed new particle formation events occurring over the GBR, and remote sensing analyses have demonstrated that AOD is positively correlated with conditions that have been empirically shown to enhance coral DMS emissions. It is therefore possible that pristine coral reefs such as the GBR are a source of MBA, with important implications for coral resilience to future rises in ocean temperature.

Corals enhance DMS emissions in response to rising stress however, emissions decline when coral physiological tolerance ranges are approached. The same trend has been observed for both AOD and LLC cover, which often coincide with coral bleaching events. Natural aerosols are an important source of CCN and cloud cover is a primary determinant of the spatial extent and severity of coral bleaching. Although the response of coral DMS emissions to the changing climate is uncertain, a decline in DMS-derived particles may exacerbate warming and the degradation of coral reefs. Capturing the complex biogeochemical and ocean-atmosphere interactions involved in the coral reef sulfur cycle is a challenging task. A multidisciplinary approach which utilises both field and remote sensing observations, with earth system models is therefore needed to improve our understanding of the importance of DMS in coral physiology and climate. This will enable a better understanding of natural aerosol radiative effects and inform alternative methods of coral reef management.

**Data availability**

The aerosol optical depth dataset presented in this review can be obtained from NASA's OceanColor Distributed Active Archive Centre (https://oceancolor.gsfc.nasa.gov/).

**Author contributions**

RJ prepared the manuscript with contributions from all co-authors.

**Competing interests**

The authors declare that they have no conflict of interest.

**Acknowledgements**

The authors gratefully acknowledge the MODIS mission scientists and NASA Ocean Biology Processing Group for the provision of aerosol optical depth data presented in this review. We also thank the NOAA Air Resources Laboratory (ARL) for the provision of the HYSPLIT transport and dispersion model and READY website (https://www.ready.noaa.gov) used in this publication.

**Funding**

Rebecca Jackson is supported by an Australian Government Research Training Program Scholarship, a Commonwealth Scientific and Industrial Research Organisation Scholarship and the Griffith University School of Environment and Science.

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

**Glossary**

Aerosol: Solid or liquid particle suspended in the atmosphere. Examples include sea spray, dust, smoke and biogenic sulfates.

Aerosol optical depth (AOD): unitless measure of the amount of aerosol in the atmosphere, determined by the extinction of solar radiation.

Albedo: the amount of solar radiation which is reflected from a surface, such as an aerosol particle or cloud droplet.

Catabolism: metabolic pathway by which compounds are degraded into smaller units. In the coral holobiont, DMSP catabolism occurs via the demethylation (removal of a methyl group) or cleavage (enzymatic splitting of a chemical bond) pathways.

Cloud condensation nuclei (CCN): aerosol particles on which water vapour condenses to form cloud droplets. CCN influence the lifetime, albedo and cover of clouds, with implications for precipitation and the Earth's radiative balance.

Coral holobiont: The coral host and all endosymbiotic micro-organisms, including microalgae, microbes and viruses.

Marine boundary layer (MBL): the lower atmosphere in contact with the ocean surface, where the exchange of heat, moisture, momentum and chemical species between the ocean and atmosphere occurs.

Radiative forcing: The net difference between top of atmosphere incoming and outgoing radiation due to interactions between solar radiation, aerosols, clouds and the Earth's surface, measured in $W\ m^{-2}$. Positive forcing indicates a warming effect, while negative forcing indicates a cooling effect.

Reactive oxygen species (ROS): reactive oxygen produced by impaired algal photosystems and coral tissues during exposure to high temperatures and/or irradiance, which damage cells and tissues. Examples include the superoxide anion, hydroxyl radical and singlet oxygen.

Solar radiation management (SRM): a solar geoengineering approach to mitigating global warming via an increase in the aerosol and cloud albedo effect.

**Figures**

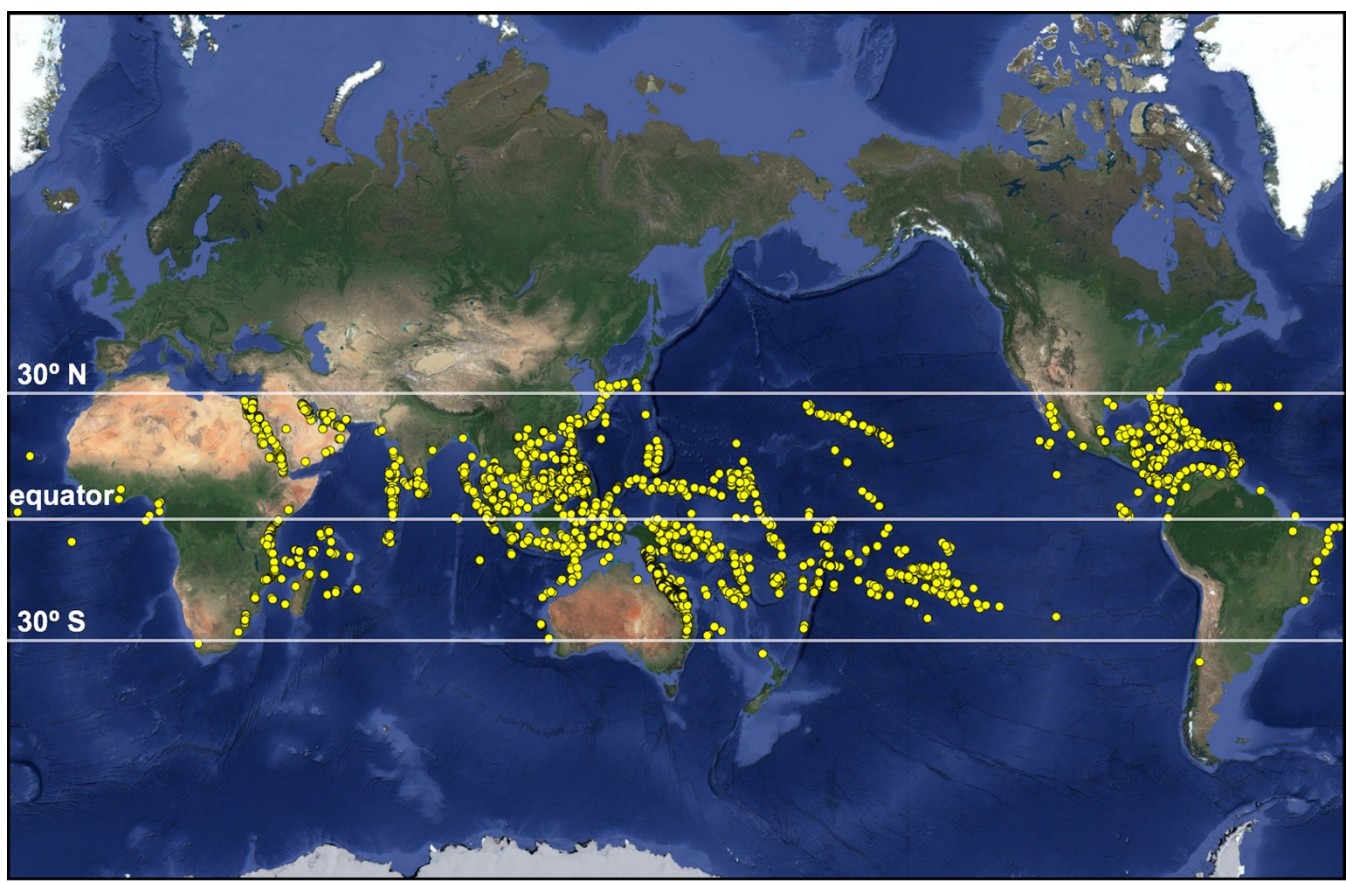

5      **Figure 1. Global distribution of tropical coral reefs (Map data ©2019 Google). Locations provided by ReefBase (http://www.reefbase.org).**

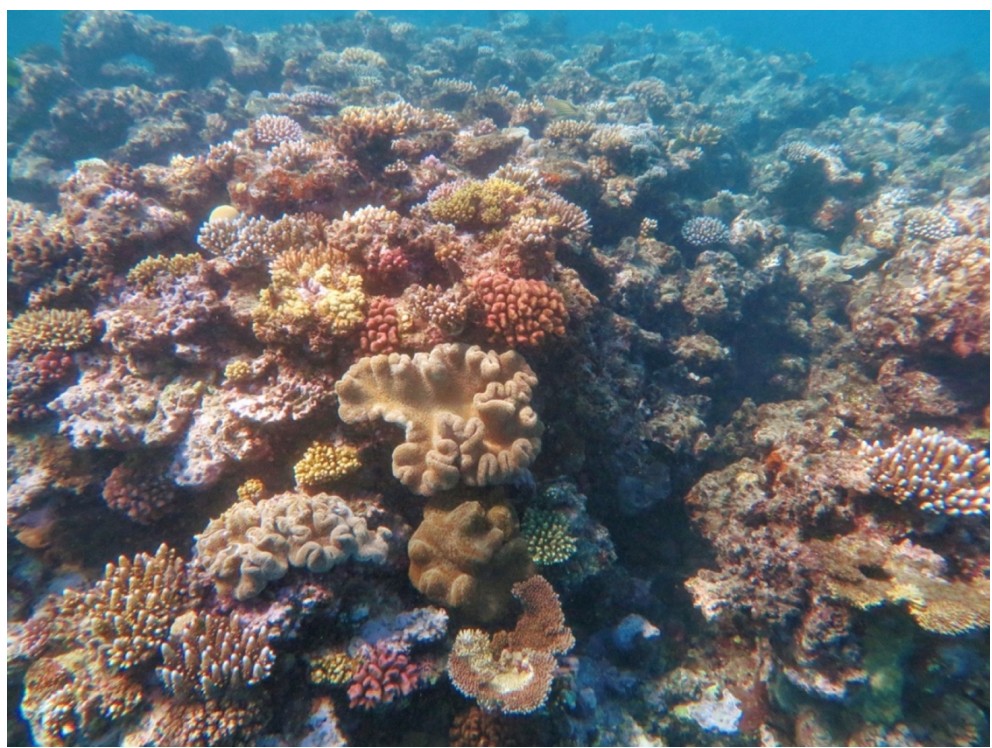

**Figure 2. Diversity of corals at Norman Reef in the northern Great Barrier Reef, Australia (R. Jackson).**

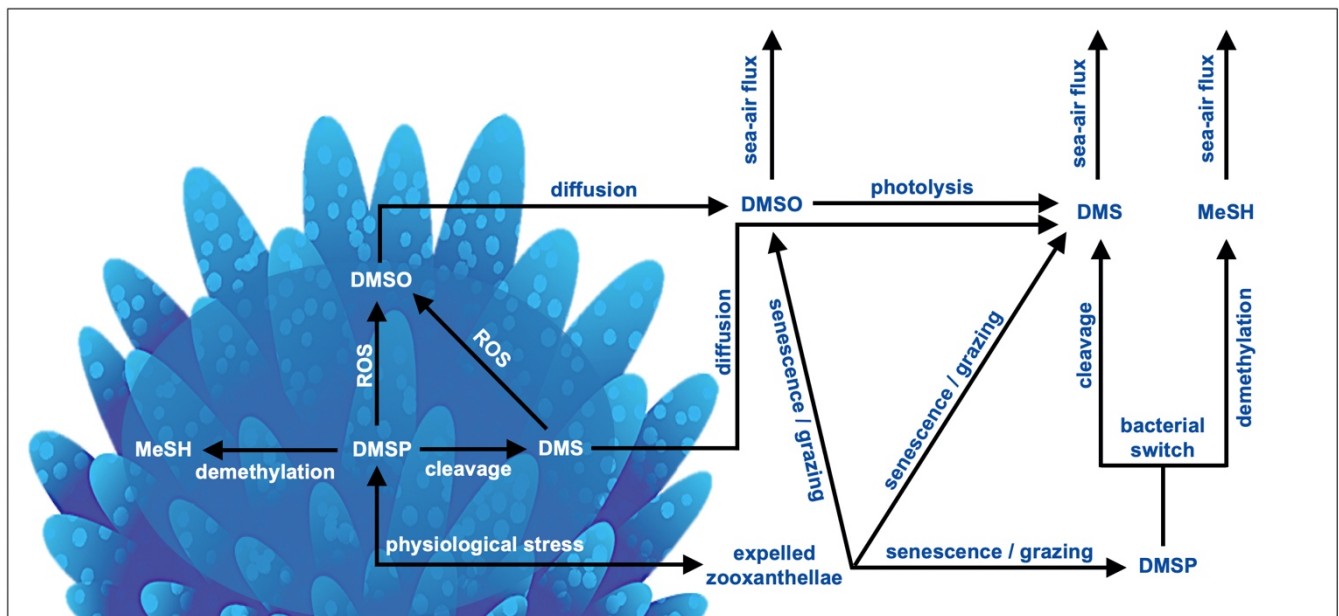

**Figure 3. Basic processes involved in the cycling of dimethylated sulfur compounds involved in a coral reef. In corals, dimethylsulfoniopropionate (DMSP) is produced by both the coral host and endosymbiotic algae in response to physiological stress. DMSP catabolism occurs via the demethylation and cleavage pathways, yielding methanethiol (MeSH) or dimethylsulfide (DMS), respectively. DMS/P scavenge reactive oxygen species (ROS) produced in coral tissues and by zooxanthellae photosystems during exposure to high temperatures, irradiance and/or hyposalinity, forming dimethyl sulfoxide (DMSO). DMS/P/O can be released to surrounding waters via zooxanthellae expulsion, followed by cell senescence and/or grazing by zooplankton or herbivorous fish. DMS/O may also diffuse from the coral into surrounding waters. Dissolved DMSP may undergo microbial catabolism, yielding organic carbon or sulfur depending on bacterial needs (bacterial switch). DMS/P/O may sink and become stored in sediment pore water. DMS/O may be ventilated to the lower atmosphere by wind and temperature-driven transfer across the ocean-atmosphere interface.**

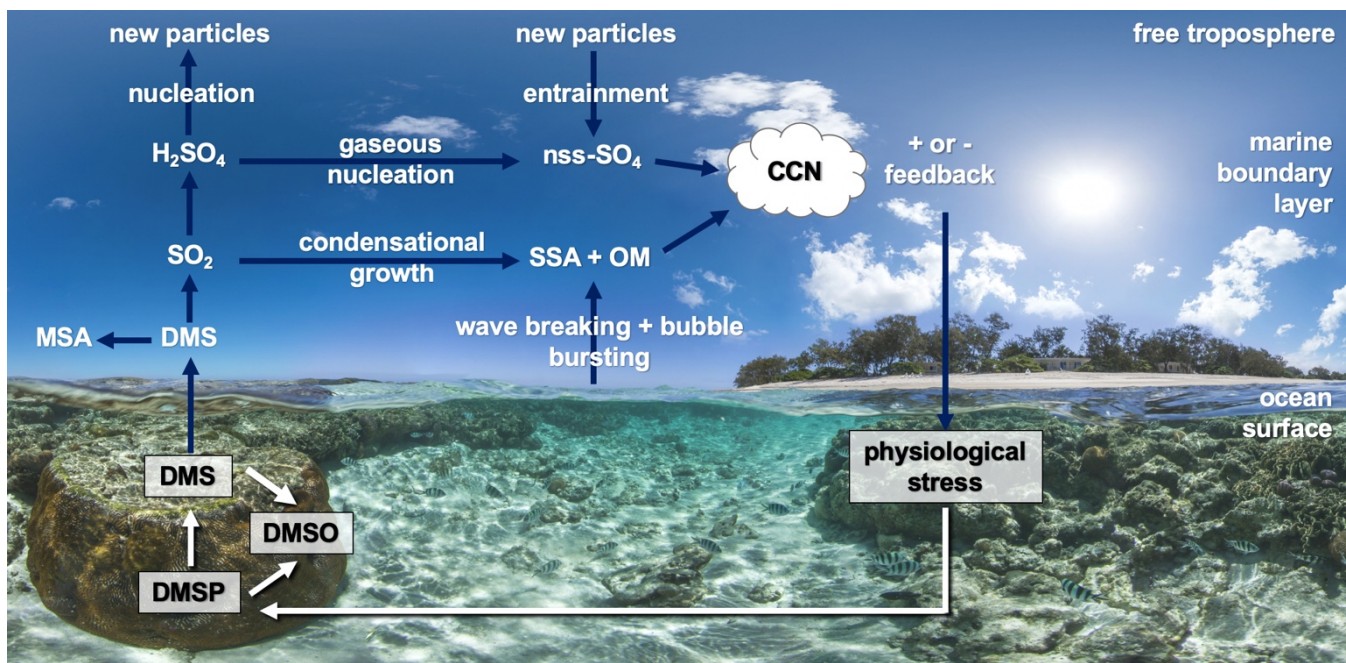

**Figure 4. Simplified overview of the coral reef dimethylsulfide (DMS)-aerosol-cloud feedback. Dimethylsulfoniopropionate (DMSP) is produced by the coral holobiont and may be oxidised to dimethyl sulfoxide (DMSO) or cleaved to DMS. Atmospheric DMS is oxidised to sulfate aerosol precursor compounds including methane sulfonic acid (MSA) and sulfur dioxide (SO$_2$) and secondarily to sulfuric acid (H$_2$SO$_4$), which may condense onto existing particles (e.g. sea-spray aerosol (SSA) or organic matter (OM)) or nucleate to non-sea salt sulfate particles (nss-SO$_4$). These may increase the number of cloud condensation nuclei (CCN) and increase cloud lifetime and albedo. Background image: http://catlinseaviewsurvey.com/gallery/i853_lady-elliot-island-australia / © Underwater Earth / XL Catlin Seaview Survey / Aaron Spence / CC BY-NC-SA 4.0.**

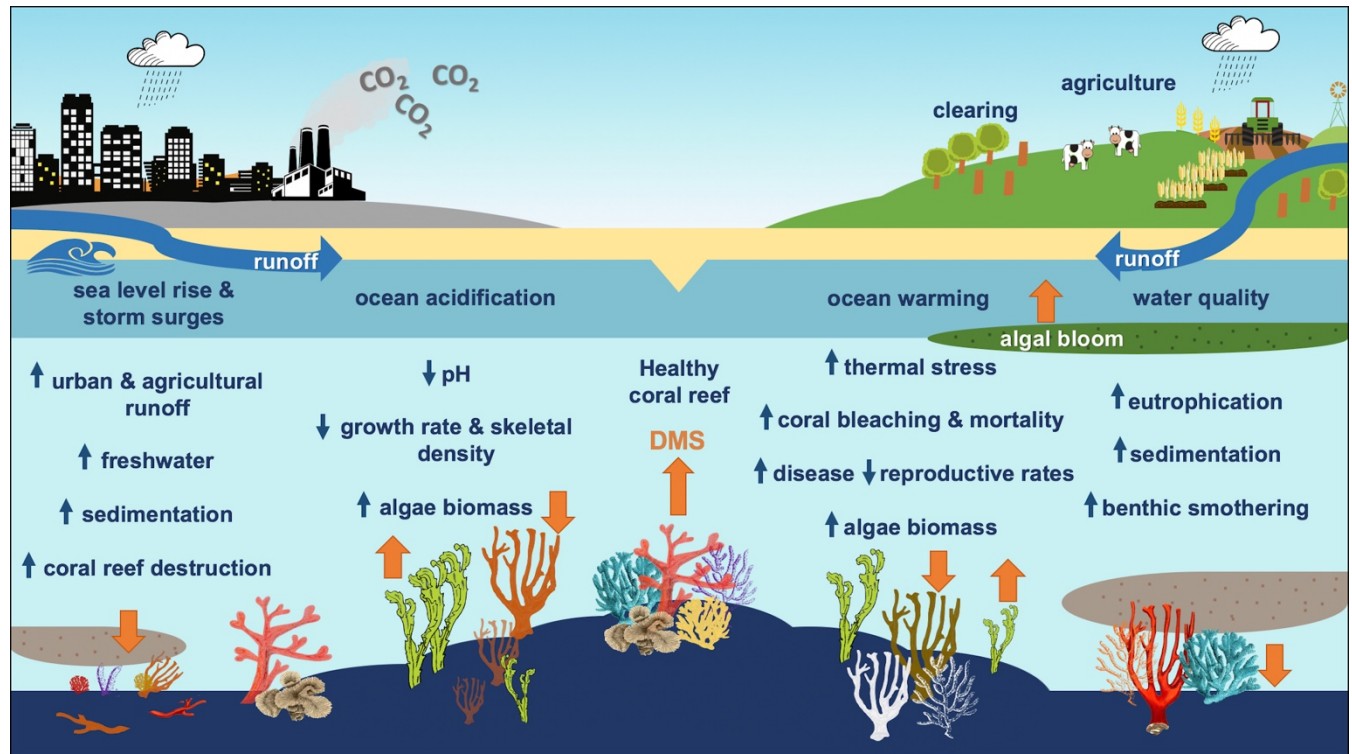

**Figure 5. The impacts of climate change and land-use on coral reefs. Orange arrows reflect the predicted increase or decrease in DMS emissions from the coral holobiont and marine micro- and macroalgae.**

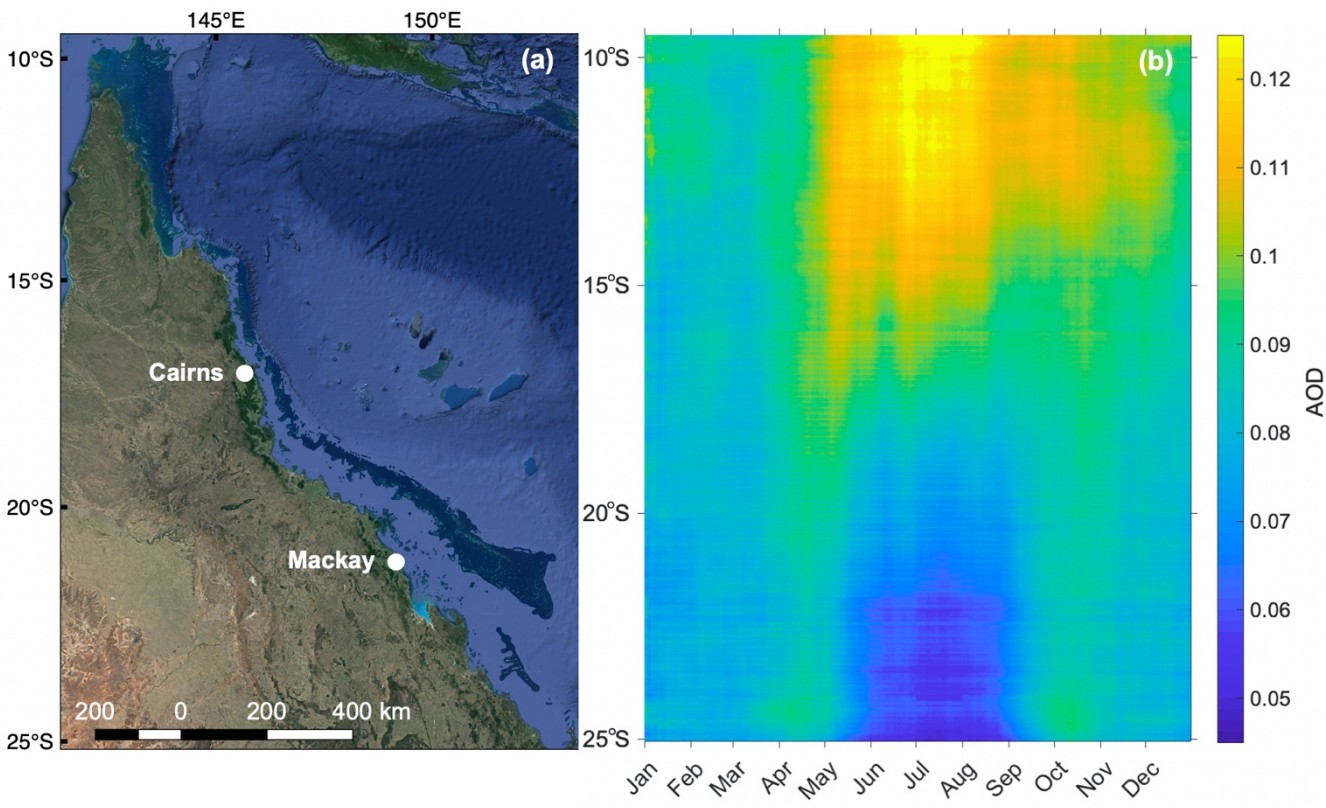

**Figure 6. (a) Map of the GBR, Australia (map data ©2019 Google), showing the region defined in (b) longitude-averaged Hövmoller plot of daily mean (2001 - 2017) total AOD (869 nm) over the GBR.**

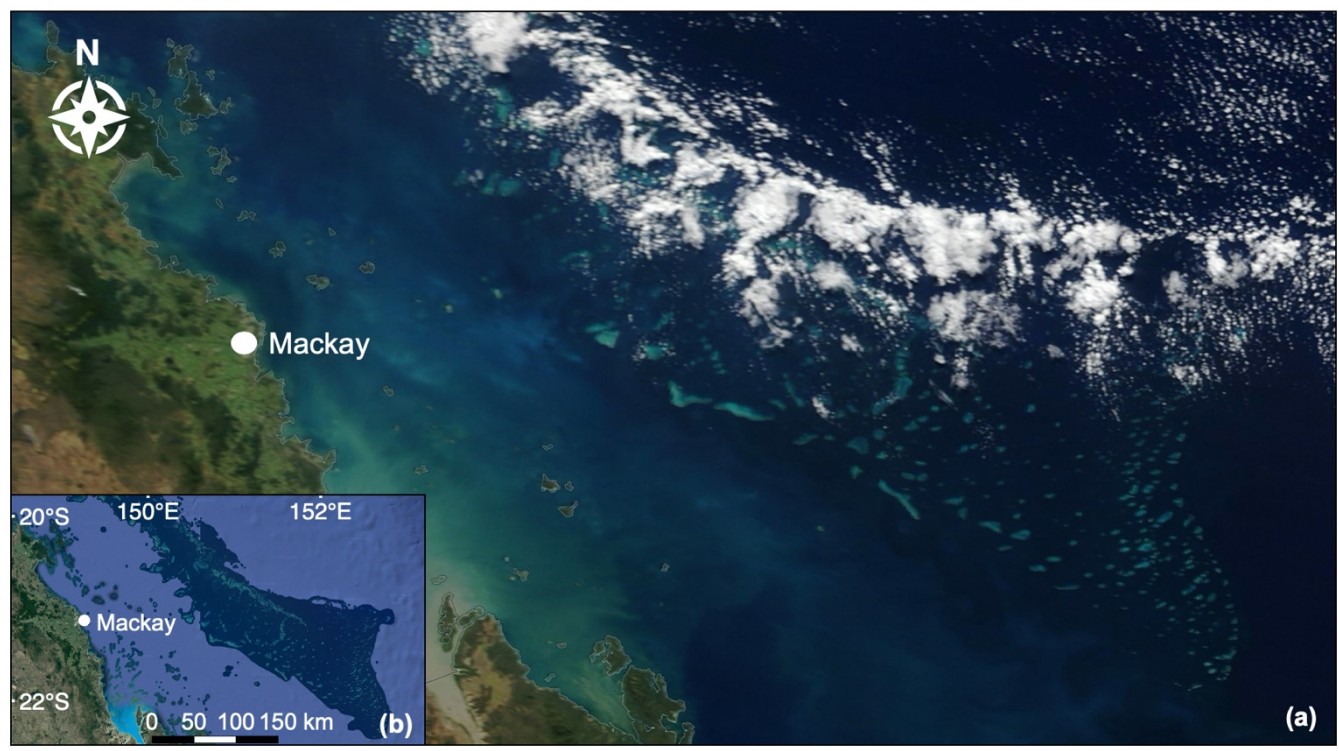

**Figure 7. (a) Low-level convective clouds aligned over the Swains Reefs in the Mackay-Capricorn Management Area of the southern GBR. True colour image captured by MODIS Terra on 31st May 2018 (NASA Earth Observing System Data and Information System). (b) Map of the southern GBR shown to be covered by clouds in the true colour image (map data ©2019 Google).**