# Peer review of "Dimethylsulfide (DMS), marine biogenic aerosols and the ecophysiology of coral reefs"

_Biogeosciences, 2019_

## Referee Comment (RC1) · Anonymous Referee #2 · 14 Oct 2019

General comments

Jackson and co-authors have produced a welcome and comprehensive review of an exciting and growing topic that bridges the gap between coral reef ecology and DMS biogeochemistry. They skilfully bring together the current threats faced by the coral reefs with a detailed overview of the role of DMS and other reduced S compounds in coral physiology, whilst placing this in the context of the intricacies of the impacts on atmospheric processes and climate regulation.

To my knowledge, a similar review has not before been published. This topic has been growing in strength over the last 10 years or so, and as such a review of this type is timely and appropriate. The manuscript is written in an accessible and easy-to-read style, with few technical or editorial issues. This paper will be of great interest to the

[Figure]

DMS community – both those with an interest in ocean biogeochemistry, as well as atmospheric chemists and modellers. It should also bring to prominence this topic amongst the coral scientific community, as until now the role of DMS/P in coral physiology, survival and its influence on the atmosphere/climate has perhaps not received the attention it deserves.

I can recommend this paper for publication provided the authors adequately address the issues which I have outlined below. In particular, I strongly recommend that the authors address my concerns with regards to sections 7 and 8 – both are currently somewhat weak and need a change in emphasis to make coral DMS production the focus of the discussion. Furthermore, both sections would benefit from being more forward-looking and include some specific recommendations for future research in the context of coral DMS production (please see further comments below).

Title: The title in its current form doesn't do a good job of describing the paper and should include some mention of DMS as this is really the main point of the paper (and more likely to get found in a Google Scholar search for 'DMS and corals'!). Perhaps something along the lines of the following: Marine dimethylsulfide (DMS) emissions and the ecophysiology of coral reefs. Dimethylsulfide (DMS) emissions from coral reefs in the face of natural and climate-induced stress.

Section 7 Coral reef monitoring: In its current form, the aim of section 7 is unclear and seems slightly weak. The authors give a reasonable overview of coral reef monitoring programs but there is little contextualisation in terms of DMS production. Some statements are unclear in their meaning e.g. L 20-22 from "Additionally, field surveys. . . ." What is meant by "the substance"? And it is not clear what is true of DMS/P in coral reef waters. The authors should revisit this section and reword to make clearer. At the moment, it feels a little like they are struggling to illustrate the relevance to DMS etc. It's also unclear what "DHW of °C-weeks" means. The final sentence of this section (L14 – 16) hints at where the authors could focus this part of the paper i.e. by providing forward-looking recommendations of future research to improve our understanding

of "the biogeochemical processes occurring in coral reefs and the ways we can effectively ensure their preservation". This is the emphasis from which they could begin this section to make it much more relevant to the review as a whole. Therefore, I recommend a re-think and re-write of this section, starting with the aim above, then drawing on the past and ongoing monitoring programs to develop recommendations for future research.

Section 8 Mitigation strategies: Similarly to section 7, the current emphasis does not seem quite relevant enough to the review as a whole. It simply serves to summarise the current literature on geoengineering etc. to protect coral reefs. Again, I believe a restructuring of this section is necessary. The authors end the section by mentioning DMS flux – I would recommend bringing this part of the discussion to the beginning of the section and then develop the more general discussion from the standpoint of DMS. Some specific, forward-looking research recommendations would also be welcomed, and this is necessary to turn what is currently a solid literature review into something more innovative.

Specific comments

Pg 4, L2: The authors refer to corals as being 'amongst the largest sources of natural sulfur' but this is incorrect. As they later explain (Pg 11) the total sea-air flux of DMS is 17.6 – 34.4 Tg S/y compared to only 0.02 – 0.08 Tg S/y from tropical coral reefs. A rewording of the sentence is required.

Pg 4, L24-28: The sentence beginning 'Particulate DMSP. . ..' would benefit from some rewording. It is currently a long sentence describing multiple phenomena and reads confusingly. The reference to grazing seems out of context here. I suggest something along these lines: 'DMSP in the form of intracellular, or particulate, DMSP (DMSPp) may be released to the surrounding reef waters via zooxanthellae expulsion at a rate of 0.2 – 0.4% Symbiodinium cells day-1, in response to elevated irradiance or temperature (ref). Furthermore, DMS or DMSO may be released in coral mucous and Symbiodinium

exudates'.

Pg 6 L23-24: This sentence isn't clear in its meaning. Please double check. Perhaps '...when ROS levels are..' could be replaced with '...by reducing ROS levels to...'.

Pg 7, L9 – 11: This is the first mention of soft corals with no previous context, and currently only offers brief information. Either add more information here, or alternatively omit soft corals from this part of the discussion, because it doesn't currently serve a great purpose.

Pg 11, 11 – 19: The findings of Six et al. (2013) (Nature CC 3, 975) and Schwinger et al. (2017) (Biogeosciences 14, 3633) should be incorporated into this part of the discussion. Also the final sentence of this paragraph is tantalising but vague and would benefit from being expanded. The relevance of carbonate chemistry and buffering capacity is currently very unclear.

Pg 12: The sub heading 4.3 Complexity of the DMS cycle doesn't quite fit. Perhaps Complexity of the climate response to DMS, or something similar.

Pg 14, L24: It may be better to say 'prevailing meteorological conditions'

Pg 16, L2: It is an overstatement to say 'If coral reefs significantly affect our climate...' perhaps say 'If coral reefs significantly affect local atmospheric conditions...' or similar.

Technical corrections

Pg 2 L10: move comma to come after '...emerging topic of research, '

Pg 6, L15: remove comma after 'approached'

Throughout the paper: The authors refer to "the radiative balance". It would be better to refer to "the Earth's radiative balance".

---

## Author Comment (AC1) · 7 Nov 2019

General comments:

The authors thank the referee for their valuable insight and comments on this discussion paper. The referee has raised important points, which have each been addressed in the revised manuscript (see supplement with tracked changed).

1. Title: The title has been changed to 'Marine biogenic aerosols, dimethylsulfide (DMS) emissions and the ecophysiology of coral reefs' to better describe the paper.

2. Section 7: We have included a substantial re-write of section 7 and have renamed this section 'Future research' to better reflect the aim of the discussion. This section now highlights gaps in the literature and presents several specific recommendations

for future research which will progress the current understanding of the role of DMS in coral reefs and local atmospheric properties (pg 16 - 18).

3. Section 8: The purpose of this section is to discuss potential ways to counteract the predicted decline in biogenic aerosol emissions with ongoing coral reef degradation and to mitigate the detrimental effects of further ocean warming. Section 8 has been re-written to focus on alternative ways to conserve coral reefs, such as the propagation of temperature tolerant coral species and climate engineering to artificially mimic marine biogenic aerosol emissions (pg 18 - 20).

Specific comments:

1. We have clarified that corals are 'amongst the largest individual sources of natural sulfur' (pg 4 L2), when compared with individual micro- or macroalgae.

2. The sentence beginning 'Particulate DMSP...' has been reworded to read more clearly (pg 4 L25 - 29).

3. The sentence reading '...when ROS levels are...' has been replaced with '...by reducing ROS levels to' (pg 6 L24).

4. More context on the importance of DMS emissions from soft corals has been provided (pg 7 L9 - 13). Although the focus of the paper is Scleractinian corals, soft corals also contain large quantities of DMS and have been reported to increase in abundance in disturbed coral reef systems.

5. The findings of Six et al. (2013) and Schwinger et al. (2017) regarding changes to DMS sea-air flux in response to ocean acidification have been incorporated into section 4.2 (pg 11, L19 - 25). The importance of regional biogeochemistry and phytoplankton community dynamics in predicting changes to DMS emissions has also been made clearer (pg 11, L25 - 29).

6. The sub-heading '4.3 Complexity of the DMS cycle' has been changed to '4.3 Complexity of the climate response to DMS' (pg 12, L12).

7. The phrase '...existing meteorological conditions...' has been changed to '...prevailing meteorological conditions' (pg 15, L9).

8. The sentence 'If coral reefs significantly affect our climate...' has been changed to 'If coral reefs significantly affect local atmospheric conditions...' (pg 16, L16).

Technical corrections:

1. Comma has been moved after '...emerging topic of research' (pg 2, L10).

2. Comma after 'approached' has been removed (pg 6, L15).

3. The phrase 'the radiative balance' has been replaced throughout the paper with 'the radiative balance over coral reefs'. We did not use 'the Earth's radiative balance' in these instances as the discussion was focused on local effects over coral reefs.

Please also note the supplement to this comment:
https://www.biogeosciences-discuss.net/bg-2019-207/bg-2019-207-AC1-supplement.pdf
* * *
[Figure]

**Supplement:**

**Reviews and syntheses: Marine biogenic aerosols, dimethylsulfide (DMS) emissions 
[revised manuscript text omitted]

---

## Referee Comment (RC2) · Michael Steinke (Referee) · 5 Feb 2020

General Comments (review prepared by Michela Lever and Michael Steinke)

Coral reefs are important sources of the climate-active trace gas dimethyl sulfide (DMS). This review summarises some of our knowledge on the impact of coral reefs on sulfur cycling and the potential role of DMS and its precursor DMSP in alleviating physiological stress in corals. The paper is generally well-written but suffers from poorly designed figures and a narrow focus on corals from the Great Barrier Reef (see below).

Specific Comments

A. The review is sometimes too narrow and focuses on Great Barrier Reef processes

only. However, there is some information available at least on Caribbean and Hawaiian reefs. For example, I am aware of the following:

Andreae, M. O., Barnard, W. R. and Ammons, J. M. (1983) The Biological Production of Dimethylsulfide in the Ocean and its Role in the Global Atmospheric Sulfur Budget. In: Hallberg, R. (ed.) Environmental Biogeochemistry Ecol. Bull. Stockholm: Ecol. Bull.

Hill, R., Li, C., Jones, A., Gunn, J. and Frade, P. (2010) Abundant betaines in reef-building corals and ecological indicators of a photoprotective role. Coral Reefs, 29, 869-880. Hill, R. W., Dacey, J. W. H. and Edward, A. (2000) Dimethylsulfoniopropionate in giant clams (Tridacnidae). Biological Bulletin, 199, 108-115.

Hill, R. W., Dacey, J. W. H. and Krupp, D. A. (1995) Dimethylsulfoniopropionate in Reef Corals. Bulletin of Marine Science, 57, 489-494.

B. Some sections only weakly link to marine biogenic aerosols (see title). Sections 7 and 8 appear to mostly cover reef conservation efforts. Possibly, Section 7 could be enhanced by adding information on atmospheric monitoring and how this could be combined with existing efforts (long-term monitoring) that quantify coral health. For example, some classic VOC studies conducted at Mace Head Observatory might be useful here:

Broadgate, W. J., Malin, G., Kupper, F. C., Thompson, A. and Liss, P. S. (2004) Isoprene and other non-methane hydrocarbons from seaweeds: a source of reactive hydrocarbons to the atmosphere. Marine Chemistry, 88, 61-73.

Carpenter, L. J., Sturges, W. T., Penkett, S. A., Liss, P. S., Alicke, B., Hebestreit, K. and Platt, U. (1999) Short-lived alkyl iodides and bromides at Mace Head, Ireland: Links to biogenic sources and halogen oxide production. Journal of Geophysical Research-Atmospheres, 104, 1679-1689.

C. Many of the figures are not well designed and/or provide little information. Figure 1

and Figure 2 can likely be removed. The caption in Figure 4 should explain all abbreviations similar to Figure 3. There are many issues with this figure, including:

a. Confusing use of terms such as 'ventilation' to explain loss of DMS to atmosphere but this is not indicated for loss of methanethiol.

b. What is meant by 'catabolism'? Is the demethylation step not a catabolism as well?

c. Why are zooplankton and phytoplankton the only sources of DMSP? Why not the coral?

d. Why is DMSP not released from other grazers such as herbivorous fish?

e. How do DMS and acrylate feed into microbial demethylation?

It is unclear how Figure 6 was generated. Where is this figure from? Is this original research and should not be presented in a 'review'? Lat/Lon and scale info is missing from Figure 6a, and why does Figure 6b display odd Latitude info? Unclear how the arrow in 6a relates to info in 6b. Is there a unit for the colour scale along the right (say on figure what it shows)? The actual reef is poorly illustrated in Figure 7. It is unclear what the white dots in the inset show, how the arrow relates between the two figures, what the dark blue area in the insert displays, etc. There is a scale and Lat/Lon information missing.

D. Some reviews contain a glossary – this may be useful here, too? Could explain specific terminology, for example: - Radiative forcing - Aerosol optical depth - . . .

Technical Corrections

Page 2, Line 1: Restructure (?): '. . .Coral reefs are being threatened by global climate change, with ocean warming, acidification and declining water quality adversely affecting coral health and cover in many coastal systems. . .'

P2, L14: '. . .Gaining a better understanding of the role of coral reef DMS emissions is crucial to predicting the future climate of our planet. . .' Is this justified? Do DMS

emissions from coral reefs really affect the climate of our planet? May be regional climate? Coastal zones? Statement here seems to fit poorly with statement on P11, L34!

P3, L6: May have to widen out to beyond the genus 'Symbiodinium' or refer to Symbiodiniaceae? See paper by LaJeunesse, T. C., Parkinson, J. E., Gabrielson, P. W., Jeong, H. J., Reimer, J. D., Voolstra, C. R. and Santos, S. R. (2018) Systematic Revision of Symbiodiniaceae Highlights the Antiquity and Diversity of Coral Endosymbionts. Current Biology, 28, 2570-2580.e6.

P3, L8: Check hyphenation: '. . .membrane-enclosed compartments...'

P4, L2: What is meant by '. . .natural sulfur . . .' – change to 'biogenic sulfur'?

P4, L7: Check hyphenation: '. . .DMS-derived aerosol . . .'

P4, L11: I think some information should be provided on conservation and management before making the statement in the final sentence of Section 1: '. . .In the face of rapid climate change, non-traditional means of conservation and management may be required. . .'

P4, L16: Reword to (?): '. . .above a coral reef exposed to air. . .'

P4, L17: Capitalisation of 'Octocorals' necessary? These are animals as in 'dogs'?

P4, L18: What are the concentrations for dinoflagellate cells and benthic algae?

P4, L21: Check structure of information. Consider removing orphan sentence/paragraph.

P4, L24-28: This sentence does not make logical sense. It starts off with information on DMSPp but then seems to include info on DMSPd (release from phytoplankton during grazing) and other non-particulate exudates.

P4, L28-29: There is also the possibility that DMS is being produced from DMSP

without the production of acrylate (ddd-cleavage pathway): Todd, J. D., Rogers, R., Li, Y. G., Wexler, M., Bond, P. L., Sun, L., Curson, A. R. J., Malin, G., Steinke, M. and Johnston, A. W. B. (2007) Structural and regulatory genes required to make the gas dimethyl sulfide in bacteria. Science, 315, 666-669. (See also P5, L4).

P5, L18: I disagree with: '…Until recently, it was thought that biosynthesis was limited to photosynthetic endosymbionts…' – (i) Many non-endosymbiotic organisms make DMSP and (ii) it has been long known that the heterotrophic dinoflagellate (= 'animal') Crypthecodinium cohnii can make DMSP, for example: Uchida, A., Ooguri, T., Ishida, T., Kitaguchi, H. and Ishida, Y. (1996) Biosynthesis of dimethylsulfoniopropionate in Crypthecodinium cohnii (Dinophyceae). In: Kiene, R. P., Visscher, P. T., Keller, M. D. and Kirst, G. O. (eds.) Biological and Environmental Chemistry of DMSP and Related Sulfonium Compounds. New York: Plenum Press.

P5, L27 (and elsewhere in text): Remove italics from 'spp.'

P6, L23: Is this true?: '…These ROS can diffuse from the algal symbiont into coral cytoplasm . . .' – I wonder whether ROS are too reactive to pass through biomembranes? Isn't this why they are so damaging to biological structures/molecules?

P6, L32: Is this statement correct? I do not recall that the paper by Hopkins et al. (2016) addresses the effects of SST or salinity.

P7, L14: Add apostrophe '…hinders corals' ability . . .'

P8, L4: What are those 'time scales'?

P8, L13: Check hyphenation '…temperature-sensitive species…'

P8, L15: Check hyphenation '…macro- and microalgae . . .'

P8, L21-22: So, this suggests that DMSP may not have a role in conferring temperature tolerance? Reconsider the wording.

P8. L25: Change singular/plural: '…However, tolerance thresholds within Symbiodinium clades are highly variable (Klueter et al., 2017) and do not always predict DMSP biosynthesis (Steinke et al. 2011)...'

P8, L26: Steinke et al. (2011) missing from reference list – carefully check all references.

P9, L2: Check hyphenation: '...algal-dominated coral-reef communities...'. Also, this sentence may require a reference,

P9, L4: Check hyphenation: '...pH-sensitive coral-calcification rates ...'.

P9, L6: Change singular/plural '...absorbed by the oceans and affect seawater ...'

P9, L7: Avoid repetitive word usage '...Increased CO2 levels increase ...'

P9, L14: It may be important to point out that this is for warm-water corals? I suspect that the damage to cold-water corals is much higher still since the dissolution of CO2 into water is enhanced/temperature dependent?

Equation 2: Should the reaction be presented the other way around, because the text refers to erosion of calcareous structures?

P9, L23: Could add effects of temperature/CO2 on DMS/P in isolation and combined? See: Arnold, H. E., Kerrison, P. and Steinke, M. (2013) Interacting effects of ocean acidification and warming on growth and DMS-production in the haptophyte coccolithophore Emiliania huxleyi. Global Change Biology, 19, 1007-1016.

P9, L30: May mention that RCP8.5 is an IPCC scenario?

P10, L24 Check hyphenation '...biologically derived negative feedback ...'

P11, L3: Change wording '...undersampled ocean regions ...'

P11, L8: It is unclear what is meant by '...all of which reduce the thermal capacity of the sea surface...'. How can the surface concentration of DMS, tidal height, etc. reduce the thermal capacity?

P11, L 9: Change wording to 'increased diffusivity'?

P11, L11: I find it difficult to see how this paragraph is relevant to the specific topic/title of manuscript – unless cold-water corals are considered here? The manuscript sometimes lacks focus.

P13, L32: More info on '...and other volatile organic compounds (VOCs) ...' would be useful here. What other gases are being released?

P14, L5: What is meant by '...tidal lunar cycle ...' – is this not the same? Lunar cycle drives the tides?

P14, L26: Avoid use of 'extreme' (it's all relative). May be 'high solar irradiance'?

P15, L8-9: Check hyphenation '...rainfall-sensitive agriculture ...'

P15, L 18: '...or condenses onto existing particles ...'

P16, L9-10: Coral bleaching is not a stressor '...impacts of environmental stressors such as coral bleaching...'. Rewording necessary.

P16, L 10: Check spacing: '...coral reefs is also...'

P16, L19: Avoid 'extreme' and check hyphenation '...are cost- and resource intensive ...'

P16, L21: What is meant by this '...Additionally, field surveys cannot capture processes that may be occurring down-wind of the substance's origin...'? Numerous atmospheric/marine chemistry studies do exactly this? And how is that true for studies on DMSP? Unclear – would need a re-write or more information?

P16, L26: Check hyphenation '....cost- and time efficient.'

P16, L31: Consider capitalisation '...Hotspot and Degree Heating Week (DHW) ...'

P16, L33: Check punctuation '...maximum, which, when accumulated over a 12-week moving window, provide a measure ...'
P17, L1: Here and elsewhere, consider presentation of water temperature/degree symbol spacing. It should be 8 °C but some journals also use 8°C (never 8° C).

P17, L20: Check hyphenation '. . .temperature-tolerant coral species . . .'

Section 8 and elsewhere: Check use of tenses. For example '. . .In a modelled scenario, injecting 5 Tg SO2 annually into the stratosphere above Caribbean coral reefs reduces SST, irradiance and sea-level rise and results in a substantial decline in the number of mass coral bleaching events predicted to occur over the next 50 years (Zhang et al., 2017). . . .'; '. . .Similarly, Kwiatkowski et al. (2015) reported that enhancing SO2 concentrations in the tropical stratosphere reduces SST and the risk of coral bleaching over the next 30 years under an RCP4.5 scenario. . .'; '. . .Latham et al. (2013) found that an enhanced source of sea-spray aerosol over the GBR, Caribbean and French Polynesia offsets the warming effects . . .' Also: consider use of 'reduce' (could be confused with chemical reduction) and use 'decrease' instead?

P18, L1 and 4: Avoid repetitive word usage 'another'.

P18, L5: Consider rewording '. . ..over reefs of high economic and environmental value.'

P18, L21: '. . .There is substantial evidence that coral reefs are strong sources of dimethylated sulfur compounds . . .' It is true that corals are producing a lot of DMSP. However, our recent attempt to simulate the DMS sea-to-air flux from coral reefs using the 'model cnidarian' Aiptasia (which likely has its limitations. . .) finds that the flux normalised to sea surface area is lower in coral reefs than the average global oceanic flux: Franchini, F. and Steinke, M. (2017) Quantification of dimethyl sulfide (DMS) production in the sea anemone Aiptasia sp. to simulate the sea-to-air flux from coral reefs. Biogeosciences, 14, 5765-5774.

P18, L 31: '. . .This biogenic aerosol source is in danger of becoming weaker with ongoing coral reef degradation. . .' Earlier, the case was made for DMS being the same/increasing when seaweeds replace corals?

References: Check for use of italics for scientific names (e.g. *Symbiodinium* in Deschaseaux et al. 2014b, Klueter et al. 2017).

---

## Author Comment (AC2) · 26 Feb 2020

The authors would like to thank the Referee for their valuable insight and constructive comments on this manuscript. Important points have been raised and we have done our best to address each of these in the revised manuscript. The Referee's comments are indicated by 'RC' below, followed by the author comments (AC). Page and line numbers in parentheses refer to the manuscript with tracked changes (see Supplement).

General comments

RC: Coral reefs are important sources of the climate-active trace gas dimethyl sulphide (DMS). This review summarises some of our knowledge on the impact of coral reefs on sulfur cycling and the potential role of DMS and its precursor DMSP in alleviating

physiological stress in corals. The paper is generally well-written but suffers from poorly designed figures and a narrow focus on corals from the Great Barrier Reef (see below).

AC: The current discussion does focus on processes in the Great Barrier Reef (GBR), because a substantial amount of the literature on biogeophysical processes relating to DMS in coral reefs focuses on this region. The GBR is considered to be relatively pristine (given the southern hemisphere location, size and predominant south-easterly trade winds), and this makes it a good study site for field and remote sensing analyses of the relationship between coral physiological stress, DMS emissions and the properties of marine aerosols and clouds, which can be confounded by continental and/or anthropogenic emissions in other regions. However, we recognise that the review is limited in scope and so have expanded the focus to include more literature from other coral reef regions. We have also made changes to figures where needed (see specific comment responses below).

Specific comments

RC: The review is sometimes too narrow and focuses on Great Barrier Reef processes only. However, there is some information available at least on Caribbean and Hawaiian reefs. For example, I am aware of the following: Andreae, M. O., Barnard, W. R. and Ammons, J. M. (1983) The Biological Production of Dimethylsulfide in the Ocean and its Role in the Global Atmospheric Sulfur Budget. In: Hallberg, R. (ed.) Environmental Biogeochemistry Ecol. Bull. Stockholm: Ecol. Bull. Hill, R., Li, C., Jones, A., Gunn, J. and Frade, P. (2010) Abundant betaines in reef building corals and ecological indicators of a photoprotective role. Coral Reefs, 29, 869-880. Hill, R.W., Dacey, J.W. H. and Edward, A. (2000) Dimethylsulfoniopropionate in giant clams (Tridacnidae). Biological Bulletin, 199, 108-115. Hill, R. W., Dacey, J. W. H. and Krupp, D. A. (1995) Dimethylsulfoniopropionate in Reef Corals. Bulletin of Marine Science, 57, 489-494.

AC: We recognise that the review has a narrow geographical focus and so have included more literature on other coral reef regions. We have made reference to the

following as suggested: Andreae et al., 1983: DMS production by corals exposed to air at low tide can dominate the background oceanic signal (page 6, line 21 and 28-32). Hill et al., 2010: Curacao reef-building corals contain an abundance of betaines, including DMSP. Biosynthesis is modulated by exposure to irradiance, indicating a potential role in photoprotection (page 7, line 1-2; page 7, line 12-14). Hill et al., 2000: Reported DMSP concentrations in giant clams (page 4, line 20-21). Hill et al., 1995: Reported DMSP concentrations measured in corals sampled from Hawaii (page 4, line 16).

RC: Some sections only weakly link to marine biogenic aerosols (see title). Sections 7 and 8 appear to mostly cover reef conservation efforts. Possibly, Section 7 could be enhanced by adding information on atmospheric monitoring and how this could be combined with existing efforts (long-term monitoring) that quantify coral health. For example, some classic VOC studies conducted at Mace Head Observatory might be useful here: Broadgate,W. J., Malin, G., Kupper, F. C., Thompson, A. and Liss, P. S. (2004) Isoprene and other non-methane hydrocarbons from seaweeds: a source of reactive hydrocarbons to the atmosphere. Marine Chemistry, 88, 61-73. Carpenter, L. J., Sturges, W. T., Penkett, S. A., Liss, P. S., Alicke, B., Hebestreit, K. and Platt, U. (1999) Short-lived alkyl iodides and bromides at Mace Head, Ireland: Links to biogenic sources and halogen oxide production. Journal of Geophysical Research-Atmospheres, 104, 1679-1689.

AC: Not all sections discuss marine biogenic aerosols and so the title has been changed to better reflect the focus of the paper: 'Dimethylsulfide (DMS) and the eco-physiology of coral reefs: implications for marine biogenic aerosols and climate'. Section 7 has been renamed 'Future research'. This section highlights gaps in the literature and provides recommendations for future research, including the importance of analysing long-term databases on coral health and atmospheric variables (e.g. AOD observations from AERONET at Lucinda in the central GBR). Such studies would progress the current understanding of the role of DMS in coral reefs and local atmospheric properties (page 17-19). Section 8 has been rewritten to focus on alternative

ways to conserve coral reefs, such as the propagation of temperature tolerant coral species and climate engineering. The purpose of section 8 is to discuss potential ways to mitigate the detrimental effects of further ocean warming in coral reefs (page 19 - 20). Broadgate et al. (2004) and Carpenter et al. (1999) both measured the concentration and flux of various VOCs produced by macroalgae in rock pools at Mace Head, Ireland. While these studies made significant contributions to the literature on biogenic trace gas emissions, the results are not directly relevant to the discussion on long-term DMS measurements from coral reefs and so, we have not made reference to these papers in section 7.

RC: Many of the figures are not well designed and/or provide little information. Figure 1 and Figure 2 can likely be removed. The caption in Figure 4 should explain all abbreviations similar to Figure 3. There are many issues with this figure, including: a. Confusing use of terms such as 'ventilation' to explain loss of DMS to atmosphere but this is not indicated for loss of methanethiol. b. What is meant by 'catabolism'? Is the demethylation step not a catabolism as well? c. Why are zooplankton and phytoplankton the only sources of DMSP? Why not the coral? d. Why is DMSP not released from other grazers such as herbivorous fish? e. How do DMS and acrylate feed into microbial demethylation? It is unclear how Figure 6 was generated. Where is this figure from? Is this original research and should not be presented in a 'review'? Lat/Lon and scale info is missing from Figure 6a, and why does Figure 6b display odd Latitude info? Unclear how the arrow in 6a relates to info in 6b. Is there a unit for the colour scale along the right (say on figure what it shows)? The actual reef is poorly illustrated in Figure 7. It is unclear what the white dots in the inset show, how the arrow relates between the two figures, what the dark blue area in the insert displays, etc. There is a scale and Lat/Lon information missing.

AC: Figure 1 shows the geographical distribution of tropical coral reefs to provide context on the abundance and thus, the importance of these ecosystems throughout the tropics. Figure 2 illustrates the diversity of a healthy coral reef. Although Fig. 1 and

Fig. 2 don't provide information additional to what has already been discussed in the text, we do think that they support the review and have included the figures in the manuscript. Figure 4 (now Figure 3) has been changed to better illustrate the processes involved in coral DMS/P/O biosynthesis and cycling. All abbreviations and processes have been defined in the caption. The term 'ventilation' has been replaced with 'sea-air flux' to indicate loss of DMS, DMSO and methanethiol to the atmosphere. Use of 'catabolism' is meant to be 'cleavage' and has been amended throughout the text and figures. Sources of DMSP illustrated in the figure and described in the caption are the coral holobiont (including the coral host and zooxanthellae), zooxanthellae expulsion followed by cell senescence and/or grazing on by zooplankton/herbivorous fish. Figure 6 presents existing MODIS data and supports the discussion in section 5.1. Although this figure has not been published elsewhere, we think that it is acceptable to include as part of a review, as other review papers have done (for example, Figure 3 in Grythe et al., 2014). The map in Fig. 6a shows the region of the GBR and Coral Sea (from Google Earth, see caption) for which the AOD Hövmoller plot in Fig. 6b was calculated for. Latitude, longitude and scale information has been added to Fig. 6a, with corresponding latitudes added to Fig. 6b. The arrow indicated that Fig. 6a represents the location of data in Fig. 6b. The arrow has been removed and this is instead explained in the caption. AOD is a unitless value and so, no unit has been included on the colour bar in Fig. 6b. Figure 7 - The 'dark blue area' in the inset (map data from Google Earth) illustrates the outline of the southern GBR, shown to be covered by clouds in the true colour image. The arrow indicated that the inset represents the region shown in the true colour image. The arrow has been removed and this is now explained in the caption. City names (represented by the white dots) have been added to all figure components. We have also added latitude, longitude and scale information to Fig. 7.

RC: Some reviews contain a glossary - this may be useful here, too? Could explain specific terminology, for example: - Radiative forcing - Aerosol optical depth - . . .

AC: We have included a glossary of key terms and phrases commonly used throughout the manuscript (page 37).

Technical corrections

RC Page 2, line 1: Restructure (?): '...Coral reefs are being threatened by global climate change, with ocean warming, acidification and declining water quality adversely affecting coral health and cover in many coastal systems...'

AC: The first sentence of the Abstract has been rephrased (page 2, line 1-2).

RC Page 2, line 14: '...Gaining a better understanding of the role of coral reef DMS emissions is crucial to predicting the future climate of our planet...' Is this justified? Do DMS emissions from coral reefs really affect the climate of our planet? May be regional climate? Coastal zones? Statement here seems to fit poorly with statement on P11, L34!

AC: We have changed the wording here to avoid overstating the importance of coral reef DMS emissions in large-scale climate: 'In the face of rapid climate change, it is crucial that we gain a better understanding of the role of DMS in local climate of coral reefs' (page 2, line 13-14).

RC Page 3, line 6: May have to widen out to beyond the genus 'Symbiodinium' or refer to Symbiodiniaceae? See paper by LaJeunesse, T. C., Parkinson, J. E., Gabrielson, P. W., Jeong, H. J., Reimer, J. D., Voolstra, C. R. and Santos, S. R. (2018) Systematic Revision of Symbiodiniaceae Highlights the Antiquity and Diversity of Coral Endosymbionts. Current Biology, 28, 2570-2580.e6.

AC: We have replaced the genus 'Symbiodinium' with 'Symbiodiniaceae' or 'zooxanthellae' throughout the manuscript to encompass the diversity of genera of endosymbiotic microalgae within corals, with reference to LaJeunesse et al., 2018 (page 3, line 6 and elsewhere).

RC Page 3, line 8: Check hyphenation: '...membrane-enclosed compartments...'

AC: Hyphen has been added to 'membrane-enclosed' (page 3, line 8-9).

RC Page 4, line 2: What is meant by '...natural sulfur...' – change to 'biogenic sulfur'?

AC: The phrase 'natural sulfur' refers to non-anthropogenic sulfur (i.e. biogenic or volcanic). Corals are amongst the strongest individual sources of biogenic sulfur and so, we have changed the wording here to 'biogenic sulfur' (page 4, line 2).

RC Page 4, line 7: Check hyphenation: '...DMS-derived aerosol...'

AC: Spaces around the hyphen in the phrase 'DMS - derived' have been removed (page 4, line 7).

RC Page 4, line 11: I think some information should be provided on conservation and management before making the statement in the final sentence of Section 1: '...In the face of rapid climate change, non-traditional means of conservation and management may be required...'

AC: We have not gone into much detail on conservation and management strategies in the introduction, as we have discussed these in section 8. This sentence has been removed. The final few sentences of the introduction now outline what will be covered in the review (page 4, line 8-11).

RC Page 4, line 16: Reword to (?): '...above a coral reef exposed to air...'

AC: We have changed the wording here to '...above a coral reef exposed to air at low tide' (page 4, line 16-17).

RC age 4, line 17: Capitalisation of 'Octocorals' necessary? These are animals as in 'dogs'?

AC: Capitalisation of 'Octocorals' has been removed (page 4, line 19).

RC Page 4, line 18: What are the concentrations for dinoflagellate cells and benthic algae?

AC: Average concentrations for benthic macroalgae and free-living microalgae sampled from the GBR are provided (page 4, line 21-22).

RC Page 4, line 21: Check structure of information. Consider removing orphan sentence/paragraph.

AC: The orphan sentence has been merged with the following paragraph (page 5, line 16-17).

RC Page 4, line 24 - 28: This sentence does not make logical sense. It starts off with information on DMSPp but then seems to include info on DMSPd (release from phytoplankton during grazing) and other non-particulate exudates.

AC: DMSPw refers to dissolved or non-particulate DMSP (rather than DMSPd). Both DMSPw and DMSPp may be released from the coral holobiont into reef waters via zooxanthellae expulsion, followed by natural senescence or grazing. This section has been rephrased to make this point clearer (page 5, line 7-9).

RC Page 4, line 28 - 29 and page 5, line 4: There is also the possibility that DMS is being produced from DMSP without the production of acrylate (ddd-cleavage pathway): Todd, J. D., Rogers, R., Li, Y. G., Wexler, M., Bond, P. L., Sun, L., Curson, A. R. J., Malin, G., Steinke, M. and Johnston, A. W. B. (2007) Structural and regulatory genes required to make the gas dimethyl sulfide in bacteria. Science, 315, 666-669. (See also P5, L4).

AC: We have described this alternate DMSP cleavage pathway in the text, with reference to Todd et al. (2007) (page 4, line 24-28).

RC Page 5, line 18: I disagree with: '. . .Until recently, it was thought that biosynthesis was limited to photosynthetic endosymbionts. . .' – (i) Many non-endosymbiotic organisms make DMSP and (ii) it has been long known that the heterotrophic dinoflagellate (= 'animal') Crypthecodinium cohnii can make DMSP, for example: Uchida, A., Ooguri, T., Ishida, T., Kitaguchi, H. and Ishida, Y. (1996) Biosynthesis of dimethylsulfoniopropionate in Crypthecodinium cohnii (Dinophyceae). In: Kiene, R. P., Visscher, P. T., Keller, M. D. and Kirst, G. O. (eds.) Biological and Environmental Chemistry of DMSP and Related Sulfonium Compounds. New York: Plenum Press.

AC: This statement refers to the coral holobiont only, for which it was thought that DMS/P biosynthesis was solely attributed to zooxanthellae. This sentence was not meant to imply that photosynthetic endosymbionts are the only source of dimethylated sulfur compounds in the ocean (we have discussed in several places that non-endosymbiotic organisms such as marine macroalgae and free-living microalgae and heterotrophs also produce DMS/P). We have rephrased this sentence to make this point clearer: 'Until recently, it was thought that biosynthesis in the coral holobiont was limited to photosynthetic endosymbionts' (page 5, line 27-28).

RC Page 5, line 27: (and elsewhere in text): Remove italics from 'spp.'

AC: Italics has been removed from 'spp.' throughout the manuscript (page 5, line 31 and elsewhere).

RC Page 6, line 23: Is this true?: '...These ROS can diffuse from the algal symbiont into coral cytoplasm...' – I wonder whether ROS are too reactive to pass through biomembranes? Isn't this why they are so damaging to biological structures/molecules?

AC: The capacity of reactive oxygen species to diffuse across biological membranes depends on pH. For example, the superoxide anion dominates in pH conditions > ~5 (as is found in the coral gastrodermis: pH ~7, Tresguerres et al., 2017) and has a limited ability to diffuse across membranes. However, in the more acidic symbiosome (pH ~4) hydroperoxyl radicals may diffuse across cell membranes and so, can diffuse from the symbiosome into the coral gastrodermis (page 7, line 6-7).

RC Page 6, line 32: Is this statement correct? I do not recall that the paper by Hopkins et al. (2016) addresses the effects of SST or salinity.

AC: Hopkins et al. (2016) was referenced here to support the statement that when corals are exposed to a high level of stress, DMS/P concentrations decline as the rate of oxidation to DMSO increases. Hopkins et al. (2016) demonstrated this when corals were exposed to air, as occurs during low tide. SST and salinity thresholds were listed as examples of conditions which can induce this antioxidant response. For clarity, we have also included 'exposure to air at low tide' (page 7, line 17).

RC Page 7, line 14: Add apostrophe '. . .hinders corals' ability. . .'

AC: An apostrophe has been added to 'corals' ability' (page 8, line 3).

RC Page 8, line 4: What are those 'time scales'?

AC: An exact time-scale can't really be provided here, given that there is significant variability in temperature and irradiance tolerance between coral species and zooxanthellae clades. In the previous sentence we state that relatively small SST anomalies can result in coral bleaching if conditions persist for 'several weeks'. We then state that SST anomalies > 2°C can cause coral bleaching in a shorter amount of time (i.e. < several weeks), depending on the magnitude of SST anomalies and duration of exposure (page 8, line 22-25).

RC Page 8, line 13: Check hyphenation '. . .temperature-sensitive species. . .'

AC: Hyphen has been added to 'temperature-sensitive' (page 9, line 2 and elsewhere).

RC Page 8, line 15: Check hyphenation '. . .macro- and microalge. . .'

AC: Hyphen has been added to 'macro- and microalgae' (page 9, line 5).

RC Page 8, line 21 - 22: So, this suggests that DMSP may not have a role in conferring temperature tolerance? Reconsider the wording.

AC: The point being made here was that temperature-tolerant zooxanthellae clades have been found to contain less DMSP compared to temperature-sensitive clades, when exposed to the same conditions. This is likely due to differences in temperature

tolerance (i.e. temperatures which induce oxidative stress in Clade C endosymbionts, may not stress Clade D, negating the need to produce as much DMS/P). We have reworded this sentence to make this point clearer (page 9, line 12-15).

RC Page 8, line 25: Change singular/plural: '...However, tolerance thresholds within Symbiodinium clades are highly variable (Klueter et al., 2017) and do not always predict DMSP biosynthesis (Steinke et al. 2011)...'

AC This sentence has been corrected (page 9, line 14-15).

RC Page 8, line 26: Steinke et al. (2011) missing from reference list – carefully check all references.

AC: We apologize for this oversight and have carefully checked that all references have been included in the reference list.

RC Page 9, line 2: Check hyphenation: '...algal-dominated coral-reef communities.... Also, this sentence may require a reference.

AC: Hyphen has been added to 'algal-dominated' (page 9, line 25). There is no need to hyphenate 'coral reef'. This sentence is a hypothetical and builds on the findings of Lin et al. (2016) who found that algae blooms can promote heat absorption at the surface (page 9, line 23-26).

RC Page 9, line 4: Check hyphenation: '...pH-sensitive coral-calcification rates...'

AC: Hyphenated the words 'pH-sensitive' (page 9, line 28). There is no need to hyphenate 'coral calcification'.

RC Page 9, line 6: Change singular/plural '...absorbed by the oceans and affect seawater...'

AC: This sentence has been corrected to 'absorbed by the oceans and affect seawater chemistry' (page 9, line 30).

RC Page 9, line 7: Avoid repetitive word usage '. . .Increased CO2 levels increase. . .'

AC: Repetition of 'increase(d)' has been removed: 'Increased CO2 levels enhance. . .' (page 9, line 31).

RC Page 9, line 14: It may be important to point out that this is for warm-water corals? I suspect that the damage to cold-water corals is much higher still since the dissolution of CO2 into water is enhanced/temperature dependent?

AC: Clarified that we are referring to projected impacts of ocean acidification in tropical coral reefs (page 10, line 5).

RC Equation 2: Should the reaction be presented the other way around, because the text refers to erosion of calcareous structures?

AC: We have reversed Eq. 2 to illustrate the process of CaCO3 erosion from left to right (page 10, line 9).

RC Page 9, line 23: Could add effects of temperature/CO2 on DMS/P in isolation and combined? See: Arnold, H. E., Kerrison, P. and Steinke, M. (2013) Interacting effects of ocean acidification and warming on growth and DMS-production in the haptophyte coccolithophore Emiliania huxleyi. Global Change Biology, 19, 1007-1016.

AC: We have discussed the effects of temperature and ocean acidification on DMS/P production in cultured zooxanthellae, highlighting the stronger influence of temperature, with reference to Arnold et al. (2013) (page 10, line 23-25).

RC Page 9, line 30: May mention that RCP8.5 is an IPCC scenario?

AC: We have included a brief definition of RCP scenarios where mentioned throughout the text (page 10, line 20-21 and page 20, line 3).

RC Page 10, line 24: Check hyphenation: '. . .biologically derived negative feedback. . .'

AC: Hyphenation of 'biologically-derived' has been removed (page 11, line 17).

RC Page 11, line 3: Change wording '...undersampled ocean regions...'

AC: The phrase 'under sampled' has been changed to 'undersampled' (page 11, line 26).

RC Page 11, line 8: It is unclear what is meant by '...all of which reduce the thermal capacity of the sea surface...'. How can the surface concentration of DMS, tidal height, etc. reduce the thermal capacity?

AC: This oversight has been corrected and now reads: 'Higher sea surface DMS concentration, SST and wind speed (which increase diffusivity), and reduced tide height, will enhance the sea-air flux of DMS in coral reef waters' (page 12, line 1-2).

RC Page 11, line 9: Change wording to 'increased diffusivity'?

AC: The phrase 'reduced diffusivity resistance' has been changed to 'increase diffusivity' (page 12, line 1).

RC Page 11, line 11: I find it difficult to see how this paragraph is relevant to the specific topic/title of manuscript – unless cold-water corals are considered here? The manuscript sometimes lacks focus.

AC: This paragraph discusses predicted changes to DMS sea-air flux from other ocean regions in a changing climate, for which the impacts in coral reefs are largely unknown. Given that the review discusses the link between DMS and atmospheric properties, we think that it is important to mention these predicted changes, to provide context on the impacts of climate change on DMS emissions and how this may influence local or regional climate (page 12, line 4-21).

RC Page 13, line 32: More info on '...and other volatile organic compounds (VOCs)...' would be useful here. What other gases are being released?

AC: Other VOCs such as isoprene may also be contributing to the seasonality observed in AOD (page 15, line 4).

RC Page 14, line 5: What is meant by '...tidal lunar cycle...' – is this not the same? Lunar cycle drives the tides?

AC: This oversight has been corrected. Use of 'tidal lunar cycle' has been replaced with 'tidal cycle' (page 15, line 10).

RC Page 14, line 26: Avoid use of 'extreme' (it's all relative). May be 'high solar irradiance'?

AC: Use of 'extreme' has been replaced with 'high' to describe solar irradiance (page 15, line 31).

RC Page 15, line 8 - 9: Check hyphenation '...rainfall-sensitive agriculture...'

AC: Hyphen has been added to the phrase 'rainfall-sensitive' (page 16, line 14).

RC Page 15, line 18: '...or condenses onto existing particles...'

AC: 'condense' has been changes to 'condenses' (page 16, line 23).

RC Page 16, line 9 - 10: Coral bleaching is not a stressor '...impacts of environmental stressors such as coral bleaching...'. Rewording necessary.

AC: Coral bleaching was mentioned here as an example of the impacts of environmental stressors, not as an example of a stressor. Section 7 has been rewritten (see general comment responses) and this sentence is no longer included (page 17-19).

RC Page 16, line 10: Check spacing: '... coral reefs is also...'

AC: Section 7 has been rewritten.

RC Page 16, line 19: Avoid 'extreme' and check hyphenation '...are cost- and resource intensive...'

AC: Replaced 'extremely' with 'very' to describe costs and time associated with field campaigns (page 18, line 10). We have hyphenated the phrase 'cost- and resource intensive' (page 18, line 10 and page 20, line 20).

RC Page 16, line 21: What is meant by this '. . .Additionally, field surveys cannot capture processes that may be occurring down-wind of the substance's origin. . .'? Numerous atmospheric/marine chemistry studies do exactly this? And how is that true for studies on DMSP? Unclear – would need a re-write or more information?

AC: This section has been rewritten: '. . .field surveys alone cannot always capture the larger scale processes involved in DMS oxidation, particle formation and growth. DMS has at atmospheric residence time of up to 1 day (Khan et al., 2016) and may therefore accumulate and affect aerosol and cloud properties over coral reefs downwind of the emissions source where field sampling may not be occurring. Therefore, it can be difficult to deduce empirical relationships between local DMS emissions and atmospheric properties from field surveys alone.' (page 18, line 14-18).

RC Page 16, line 26: Check hyphenation '. . .cost- and time efficient.'

AC: Again, the word 'cost' in 'cost- and time efficient' has been hyphenated (page 18, line 10 and page 20, line 20).

RC Page 16, line 31: Consider capitalisation '. . .Hotspot and Degree Heating Week (DHW). . .'

AC: Degree Heating Week and Hotspot have been capitalised (page 19, line 1).

RC Page 16, line 33: Check punctuation '. . .maximum, which, when accumulated over a 12-week moving window, provide a measure. . .'

AC: This sentence has been removed from the updated manuscript.

RC Page 17, line 1: Here and elsewhere, consider presentation of water temperature/degree symbol spacing. It should be 8 °C but some journals also use 8°C (never 8° C).

AC: The inclusion of a space between the degree symbol and 'C' was a formatting error which occurred when inserting a symbol in MS Word. We had fixed some of these

errors prior to submitting the manuscript. We thank you for pointing out the remaining errors which were not picked up in our revisions (page 6, line 5 and elsewhere).

RC Page 17, line 20: Check hyphenation '...temperature-tolerant coral species...'

AC: Hyphenated 'temperature-tolerant' (page 17, line 27)

RC Section 8 and elsewhere: Check use of tenses. For example '...In a modelled scenario, injecting 5 Tg SO2 annually into the stratosphere above Caribbean coral reefs reduces SST, irradiance and sea-level rise and results in a substantial decline in the number of mass coral bleaching events predicted to occur over the next 50 years (Zhang et al., 2017)...'; '...Similarly, Kwiatkowski et al. (2015) reported that enhancing SO2 concentrations in the tropical stratosphere reduces SST and the risk of coral bleaching over the next 30 years under an RCP4.5 scenario...'; ...Latham et al. (2013) found that an enhanced source of sea-spray aerosol over the GBR, Caribbean and French Polynesia offsets the warming effects...' Also: consider use of 'reduce' (could be confused with chemical reduction) and use 'decrease' instead?

AC: Past tense has been used here and elsewhere when discussing results from past publications (section 8, page 19-20).

RC Page 18, line 1 and 4: Avoid repetitive word usage 'another'

AC: Removed repetition of 'another' (page 20, line 12).

RC Page 18, line 5: Consider rewording '...over reefs of high economic and environmental value.'

AC: Reworded this sentence to '...over coral reefs with high economic and environmental value' (page 20, line 14).

RC Page 18, line 21: '...There is substantial evidence that coral reefs are strong sources of dimethylated sulfur compounds...' It is true that corals are producing a lot of DMSP. However, our recent attempt to simulate the DMS sea-to-air flux from coral

reefs using the 'model cnidarian' Aiptasia (which likely has its limitations. . .) finds that the flux normalised to sea surface area is lower in coral reefs than the average global oceanic flux: Franchini, F. and Steinke, M. (2017) Quantification of dimethyl sulfide (DMS) production in the sea anemone Aiptasia sp. to simulate the sea-to-air flux from coral reefs. Biogeosciences, 14, 5765-5774.

AC: DMS emissions from the GBR are typically in the range expected for other ocean regions ($\sim$100-200 ppt). However, several studies have also found that Acropora spp. are very strong producers of DMS/P, with concentrations > 3500 nmol cm-2 coral surface (Broadbent et al., 2002). The concentrations reported for Aiptasia in Franchini and Steinke (2017) were relatively low. It would be very interesting to investigate the contribution of coral reefs to global DMS sea-air flux estimates using concentrations reported for other cnidarians, particularly Acropora corals. Acropora are abundant throughout the Indo-Pacific and likely dominate the coral source of DMS/P in this region (Swan et al., 2017b). Concentrations of atmospheric DMS above the Heron Island coral reef flat reached $\sim$45 nmol m-3 (> 1000 ppt) when the corals were exposed to air at low tide (Swan et al., 2017a). This suggests that the sea-air flux of DMS from corals can be a much higher, albeit intermittent, source of DMS than the oceanic signal which is dominated by algal emissions. Other studies estimate that the GBR emits 0.02 - 0.08 Tg S yr-1. Although this represents < 1% of global DMS sea-air flux estimates, it is still a significant amount of sulfur released from a relatively small region of the ocean, with the potential to influence local or possibly regional climate.

RC Page 18, line 31: '. . .This biogenic aerosol source is in danger of becoming weaker with ongoing coral reef degradation. . .' Earlier, the case was made for DMS being the same/increasing when seaweeds replace corals?

AC: The point being made here was that ongoing coral reef degradation might lead to the loss of a coral-only source of MBA (i.e. not including other potential sources such as marine algae). We can see that this point was not clear and so have removed this sentence to avoid confusion (page 21, line 9).

RC References: Check for use of italics for scientific names (e.g. *Symbiodinium* in Deschaseaux et al. 2014b, Klueter et al. 2017).

AC: Scientific names have been italicised in the reference list where required.

Please also note the supplement to this comment:
https://www.biogeosciences-discuss.net/bg-2019-207/bg-2019-207-AC2-supplement.pdf

**Supplement:**

**Reviews and syntheses: Dimethylsulfide (DMS) and the ecophysiology of coral reefs: implications for marine biogenic aerosols and climate.**

[revised manuscript text omitted]

---

## Author Response (AR1)

**Author responses**

We would like to thank the Associate Editor Dr Wajih Naqvi for their efforts in facilitating the review of this manuscript. We particularly thank both Referees for their constructive comments, which have greatly enhanced the manuscript. Specific responses to each Referee comment appears in non-underlined text below, with line numbers in parentheses referring to the revised manuscript.

**Response to Referee #2**

**General comments**

Jackson and co-authors have produced a welcome and comprehensive review of an exciting and growing topic that bridges the gap between coral reef ecology and DMS biogeochemistry. They skilfully bring together the current threats faced by the coral reefs with a detailed overview of the role of DMS and other reduced S compounds in coral physiology, whilst placing this in the context of the intricacies of the impacts on atmospheric processes and climate regulation.

To my knowledge, a similar review has not before been published. This topic has been growing in strength over the last 10 years or so, and as such a review of this type is timely and appropriate. The manuscript is written in an accessible and easy-to-read style, with few technical or editorial issues. This paper will be of great interest to the DMS community – both those with an interest in ocean biogeochemistry, as well as atmospheric chemists and modellers. It should also bring to prominence this topic amongst the coral scientific community, as until now the role of DMS/P in coral physiology, survival and its influence on the atmosphere/climate has perhaps not received the attention it deserves.

I can recommend this paper for publication provided the authors adequately address the issues which I have outlined below. In particular, I strongly recommend that the authors address my concerns with regards to sections 7 and 8 – both are currently somewhat weak and need a change in emphasis to make coral DMS production the focus of the discussion. Furthermore, both sections would benefit from being more forward-looking and include some specific recommendations for future research in the context of coral DMS production (please see further comments below).

Title: The title in its current form doesn't do a good job of describing the paper and should include some mention of DMS as this is really the main point of the paper (and more likely to get found in a Google Scholar search for 'DMS and corals'!). Perhaps something along the lines of the following: Marine dimethylsulfide (DMS) emissions and the ecophysiology of coral reefs. Dimethylsulfide (DMS) emissions from coral reefs in the face of natural and climate-induced stress.

Section 7 Coral reef monitoring: In its current form, the aim of section 7 is unclear and seems slightly weak. The authors give a reasonable overview of coral reef monitoring programs but there is little contextualisation in terms of DMS production. Some statements are unclear in their meaning e.g. L 20-22 from "Additionally, field surveys….". What is meant by "the substance"? And it is not clear what is true of DMS/P in coral reef waters. The authors should revisit this section and reword to make clearer. At the moment, it feels a little like they are struggling to illustrate the relevance to DMS etc. It's also unclear what "DHW of °C-weeks" means. The final sentence of this section (L14 – 16) hints at where the authors could focus this part of the paper i.e. by providing forward-looking recommendations of future research to improve our understanding of "the biogeochemical processes occurring in coral reefs and the ways we can effectively ensure their preservation". This is the emphasis from which they could begin this section to make it much more relevant to the review as a whole. Therefore, I recommend a re-think and re-write of this section, starting with the aim above, then drawing on the past and ongoing monitoring programs to develop recommendations for future research.

Section 8 Mitigation strategies: Similarly to section 7, the current emphasis does not seem quite relevant enough to the review as a whole. It simply serves to summarise the current literature on geoengineering etc. to protect coral reefs. Again, I believe a restructuring of this section is necessary. The authors end the section by mentioning DMS flux – I would recommend bringing this part of the discussion to the beginning of the section and then develop the more general discussion from the standpoint of DMS. Some specific, forward-looking research recommendations would also be welcomed, and this is necessary to turn what is currently a solid literature review into something more innovative.

We thank the referee for their valuable insight and comments on our manuscript. Important points have been raised, which have been addressed in the revised manuscript.

Title: The title has been changed to 'Dimethylsulfide (DMS), marine biogenic aerosols and the ecophysiology of coral reefs' to better describe the paper.

Section 7: We have included a substantial re-write of section 7 and have renamed this section 'Future research' to better reflect the aim of the discussion. This section now highlights gaps in the literature and presents several specific recommendations for future research which will progress the current understanding of the role of DMS in coral reefs and local atmospheric properties (page 17-19).

Section 8: The purpose of this section is to discuss potential ways to counteract the predicted decline in biogenic aerosol emissions with ongoing coral reef degradation and to mitigate the detrimental effects of further ocean warming. Section 8 has been re-written to focus on alternative ways to conserve coral reefs, such as the propagation of temperature tolerant coral species and climate engineering to artificially mimic marine biogenic aerosol emissions (page 19-21).

**Specific comments**

Pg 4, L2: The authors refer to corals as being 'amongst the largest sources of natural sulfur' but this is incorrect. As they later explain (Pg 11) the total sea-air flux of DMS is 17.6 – 34.4 Tg S/y compared to only 0.02 – 0.08 Tg S/y from tropical coral reefs. A rewording of the sentence is required.

We have clarified that corals are 'amongst the largest *individual* sources of natural sulfur' (page 2, line 3 and page 4, line 2), when compared with individual micro- or macroalgae (page 4, line 21-23).

Pg 4, L24-28: The sentence beginning 'Particulate DMSP...' would benefit from some rewording. It is currently a long sentence describing multiple phenomena and reads confusingly. The reference to grazing seems out of context here. I suggest something along these lines: 'DMSP in the form of intracellular, or particulate, DMSP (DMSPp) may be released to the surrounding reef waters via zooxanthellae expulsion at a rate of 0.2 – 0.4%

Symbiodinium cells day-1, in response to elevated irradiance or temperature (ref). Furthermore, DMS or DMSO may be released in coral mucous and Symbiodinium exudates'.

This section has been reworded to read more clearly (page 5, line 7-12).

Pg 6 L23-24: This sentence isn't clear in its meaning. Please double check. Perhaps '…when ROS levels are..' could be replaced with '…by reducing ROS levels to…'.

The sentence reading '…when ROS levels are…' has been replaced with '…by reducing ROS levels to' (page 7, line 7).

Pg 7, L9 – 11: This is the first mention of soft corals with no previous context, and currently only offers brief information. Either add more information here, or alternatively omit soft corals from this part of the discussion, because it doesn't currently serve a great purpose.

More context on the importance of DMS emissions from soft corals has been provided (page 4, line 19-21 and page 7, line 28-31). Although the focus of the paper is Scleractinian corals, soft corals also contain large quantities of DMS and have been reported to increase in abundance in disturbed coral reef systems.

Pg 11, 11 – 19: The findings of Six et al. (2013) (Nature CC 3, 975) and Schwinger et al. (2017) (Biogeosciences 14, 3633) should be incorporated into this part of the discussion. Also the final sentence of this paragraph is tantalising but vague and would benefit from being expanded. The relevance of carbonate chemistry and buffering capacity is currently very unclear.

The findings of Six et al. (2013) and Schwinger et al. (2017) regarding changes to DMS sea-air flux in response to ocean acidification have been incorporated into section 4.2 (page 12, line 11-17). The importance of regional biogeochemistry and phytoplankton community dynamics in predicting changes to DMS emissions has also been made clearer (page 12, line 17-19).

Pg 12: The sub heading 4.3 Complexity of the DMS cycle doesn't quite fit. Perhaps Complexity of the climate response to DMS, or something similar.

The sub-heading has been changed to '4.3 Complexity of the climate response to DMS' (page 13, line 5).

Pg 14, L24: It may be better to say 'prevailing meteorological conditions'.

This phrase has been changed to '…prevailing meteorological conditions' (page 16, line 2).

Pg 16, L2: It is an overstatement to say 'If coral reefs significantly affect our climate…' perhaps say 'If coral reefs significantly affect local atmospheric conditions…' or similar.

The sentence 'If coral reefs significantly affect our climate…' has been changed to 'If coral reefs significantly affect local atmospheric conditions…' (page 17, line 11).

**Technical corrections**

Pg 2 L10: move comma to come after '…emerging topic of research, '

Comma has been moved *after* '…emerging topic of research' (page 2, line 10).

Pg 6, L15: remove comma after 'approached'

Comma after 'approached' has been removed (page 6, line 24).

Throughout the paper: The authors refer to "the radiative balance". It would be better to refer to "the Earth's radiative balance".

The phrase 'the radiative balance' has been replaced throughout the paper with 'the radiative balance over the ocean/coral reefs' (page 2, line 5-6; page 20, line 31-32; page 21, line 7 and elsewhere). We did not use 'the Earth's radiative balance' in these instances as the discussion was focused on local effects over coral reefs.

**Response to Referee #3**

**General comments**

We would like to thank the Referee for their valuable insight and constructive comments on this manuscript. Important points have been raised and we have done our best to address each of these in the revised manuscript.

**General comments**

Coral reefs are important sources of the climate-active trace gas dimethyl sulphide (DMS). This review summarises some of our knowledge on the impact of coral reefs on sulfur cycling and the potential role of DMS and its precursor DMSP in alleviating physiological stress in corals. The paper is generally well-written but suffers from poorly designed figures and a narrow focus on corals from the Great Barrier Reef (see below).

The current discussion does focus on processes in the Great Barrier Reef (GBR), because a substantial amount of the literature on biogeophysical processes relating to DMS in coral reefs focuses on this region. The GBR is considered to be relatively pristine (given the southern hemisphere location, size and predominant south-easterly trade winds), and this makes it a good study site for field and remote sensing analyses of the relationship between coral physiological stress, DMS emissions and the properties of marine aerosols and clouds, which can be confounded by continental and/or anthropogenic emissions in other regions (page 13, line 30-page 14, line 2). However, we recognise that the review is limited in scope and so have expanded the focus to include more literature from other coral reef regions. We have also made changes to figures where needed (see specific comment responses below).

**Specific comments**

The review is sometimes too narrow and focuses on Great Barrier Reef processes only. However, there is some information available at least on Caribbean and Hawaiian reefs. For example, I am aware of the following:

Andreae, M. O., Barnard, W. R. and Ammons, J. M. (1983) The Biological Production of Dimethylsulfide in the Ocean and its Role in the Global Atmospheric Sulfur Budget. In: Hallberg, R. (ed.) Environmental Biogeochemistry Ecol. Bull. Stockholm: Ecol. Bull.

Hill, R., Li, C., Jones, A., Gunn, J. and Frade, P. (2010) Abundant betaines in reef building corals and ecological indicators of a photoprotective role. Coral Reefs, 29, 869-880.

Hill, R.W., Dacey, J.W. H. and Edward, A. (2000) Dimethylsulfoniopropionate in giant clams (Tridacnidae). Biological Bulletin, 199, 108-115.

Hill, R. W., Dacey, J. W. H. and Krupp, D. A. (1995) Dimethylsulfoniopropionate in Reef Corals. Bulletin of Marine Science, 57, 489-494.

We recognise that the review has a narrow geographical focus and so have included more literature on other coral reef regions. We have made reference to the following as suggested:

- Andreae et al., 1983 - DMS production by corals exposed to air at low tide can dominate the background oceanic signal (page 6, line 24-28).
- Hill et al., 2010 - Curacao reef-building corals contain an abundance of betaines, including DMSP. Biosynthesis is modulated by exposure to irradiance, indicating a potential role in photoprotection (page 7, line 1-2 and page 7, line 12 - 14).
- Hill et al., 2000 - Reported DMSP concentrations in giant clams (page 4, line 19-21).
- Hill et al., 1995 - Reported DMSP concentrations measured in corals sampled from Hawaii (page 4, line 15-16).

Some sections only weakly link to marine biogenic aerosols (see title). Sections 7 and 8 appear to mostly cover reef conservation efforts. Possibly, Section 7 could be enhanced by adding information on atmospheric monitoring and how this could be combined with existing efforts (long-term monitoring) that quantify coral health. For example, some classic VOC studies conducted at Mace Head Observatory might be useful here:

Broadgate,W. J., Malin, G., Kupper, F. C., Thompson, A. and Liss, P. S. (2004) Isoprene and other non-methane hydrocarbons from seaweeds: a source of reactive hydrocarbons to the atmosphere. Marine Chemistry, 88, 61-73.

Carpenter, L. J., Sturges, W. T., Penkett, S. A., Liss, P. S., Alicke, B., Hebestreit, K. and Platt, U. (1999) Short-lived alkyl iodides and bromides at Mace Head, Ireland: Links to biogenic sources and halogen oxide production. Journal of Geophysical Research-Atmospheres, 104, 1679-1689.

Not all sections discuss marine biogenic aerosols and so the title has been changed to better reflect the focus of the paper: 'Dimethylsulfide (DMS), marine biogenic aerosols and the ecophysiology of coral reefs'.

Section 7 has been renamed 'Future research'. This section highlights gaps in the literature and provides recommendations for future research, including the importance of analysing long-term databases on coral health and atmospheric variables (e.g. AOD observations from AERONET at Lucinda in the central GBR). Such studies would progress the current understanding of the role of DMS in coral reefs and local atmospheric properties (page 17-19). Broadgate et al. (2004) and Carpenter et al. (1999) both measured the concentration and flux of various VOCs produced by macroalgae in rock pools at Mace Head, Ireland. While these studies made significant contributions to the literature on biogenic trace gas emissions, the results are not directly relevant to the discussion on long-term DMS measurements from coral reefs and so, we have not made reference to these papers in section 7.

Section 8 has been rewritten to focus on alternative ways to conserve coral reefs, such as the propagation of temperature tolerant coral species and climate engineering. The purpose of section 8 is to discuss potential ways to counteract the predicted decline in biogenic aerosol emissions with ongoing coral reef degradation and to mitigate the detrimental effects of further ocean warming (page 19-21).

Many of the figures are not well designed and/or provide little information. Figure 1 and Figure 2 can likely be removed. The caption in Figure 4 should explain all abbreviations similar to Figure 3. There are many issues with this figure, including:

a. Confusing use of terms such as 'ventilation' to explain loss of DMS to atmosphere but this is not indicated for loss of methanethiol.

b. What is meant by 'catabolism'? Is the demethylation step not a catabolism as well?

c. Why are zooplankton and phytoplankton the only sources of DMSP? Why not the coral?

d. Why is DMSP not released from other grazers such as herbivorous fish?

e. How do DMS and acrylate feed into microbial demethylation? It is unclear how Figure 6 was generated. Where is this figure from? Is this original research and should not be presented in a 'review'? Lat/Lon and scale info is missing from Figure 6a, and why does Figure 6b display odd Latitude info? Unclear how the arrow in 6a relates to info in 6b. Is there a unit for the colour scale along the right (say on figure what it shows)? The actual reef is poorly illustrated in Figure 7. It is unclear what the white dots in the inset show, how the arrow relates between the two figures, what the dark blue area in the insert displays, etc. There is a scale and Lat/Lon information missing.

Figure 1 shows the geographical distribution of tropical coral reefs to provide context on the abundance and thus, the importance of these ecosystems throughout the tropics. Figure 2 illustrates the diversity of a healthy coral reef. Although Fig. 1 and Fig. 2 don't provide information additional to what has already been discussed in the text, we do think that they support the review and so, have included the figures in the manuscript.

Figure 4 (now Figure 3) has been changed to better reflect the processes involved in coral DMS/P/O biosynthesis and cycling. All abbreviations and processes have been defined in the caption. The term 'ventilation' has been replaced with 'sea-air flux' to indicate loss of DMS, DMSO and methanethiol to the atmosphere. Use of 'catabolism' is meant to be 'cleavage' and has been amended throughout the text and figures. Sources of DMSP illustrated in the figure and described in the caption are the coral holobiont (including the coral host and zooxanthellae), zooxanthellae expulsion followed by cell senescence and/or grazing on by zooplankton/herbivorous fish.

Figure 6 presents existing MODIS data and supports the discussion in section 5.1. Although this figure has not been published elsewhere, we do think that it is acceptable to include as part of a review, as other review papers have done (for example, Figure 3 in Grythe et al., 2014). The map in Fig. 6a shows the region of the GBR and Coral Sea (from Google Earth, see caption) for which the AOD Hövmoller plot in Fig. 6b was calculated for. Latitude, longitude and scale information has been added to Fig. 6a, with corresponding latitudes added to Fig. 6b. The arrow indicated that Fig. 6a represents the location of data in Fig. 6b. The arrow has been removed and this is instead explained in the caption. AOD is a unitless value and so, no unit has been included on the colour bar in Fig. 6b.

Figure 7 - The 'dark blue area' in the inset (map data from Google Earth) illustrates the outline of the southern GBR, shown to be covered by clouds in the true colour image. The arrow indicated that the inset represents the region shown in the true colour image. The arrow has been removed and this is now explained in the caption. City names (represented by the white dots) have been added to all figure components. We have also added latitude, longitude and scale information to Fig. 7.

Some reviews contain a glossary - this may be useful here, too? Could explain specific terminology, for example: - Radiative forcing - Aerosol optical depth - …

We have included a glossary of key terms and phrases commonly used throughout the manuscript (page 37).

**Technical corrections**

Page 2, line 1: Restructure (?): '…Coral reefs are being threatened by global climate change, with ocean warming, acidification and declining water quality adversely affecting coral health and cover in many coastal systems…'

The first sentence of the Abstract has been rephrased (page 2, line 1 - 2).

Page 2, line 14: '…Gaining a better understanding of the role of coral reef DMS emissions is crucial to predicting the future climate of our planet…' Is this justified? Do DMS emissions from coral reefs really affect the climate of our planet? May be regional climate? Coastal zones? Statement here seems to fit poorly with statement on P11, L34!

We have changed the wording here to avoid overstating the importance of coral reef DMS emissions in large-scale climate: '…it is crucial that we gain a better understanding of the role of DMS in local climate of coral reefs' (page 2, line 13-14).

Page 3, line 6: May have to widen out to beyond the genus 'Symbiodinium' or refer to Symbiodiniaceae? See paper by LaJeunesse, T. C., Parkinson, J. E., Gabrielson, P. W., Jeong, H. J., Reimer, J. D., Voolstra, C. R. and Santos, S. R. (2018) Systematic Revision of

Symbiodiniaceae Highlights the Antiquity and Diversity of Coral Endosymbionts. Current Biology, 28, 2570-2580.e6.

We have replaced the genus 'Symbiodinium' with 'Symbiodiniaceae' or 'zooxanthellae' throughout the manuscript to encompass the diversity of genera of endosymbiotic microalgae within corals, with reference to LaJeunesse et al., 2018 (page 3, line 6, 7 and elsewhere).

Page 3, line 8: Check hyphenation: '…membrane-enclosed compartments...'

Hyphen has been added to 'membrane-enclosed' (page 3, line 8-9).

Page 4, line 2: What is meant by '…natural sulfur…' – change to 'biogenic sulfur'?

The phrase 'natural sulfur' refers to non-anthropogenic sulfur (i.e. biogenic or volcanic). Corals are amongst the strongest individual sources of biogenic sulfur and so, we have changed the wording here to 'biogenic sulfur' (page 4, line 2).

Page 4, line 7: Check hyphenation: '…DMS-derived aerosol…'

Spaces around the hyphen in the phrase 'DMS - derived' have been removed (page 4, line 7).

Page 4, line 11: I think some information should be provided on conservation and management before making the statement in the final sentence of Section 1: '…In the face of rapid climate change, non-traditional means of conservation and management may be required…'

We have not gone into much detail on conservation and management strategies in the introduction, as we have discussed these in section 8. This sentence has been removed. The final few sentences of the introduction now outline what will be covered in the review (page 4, line 8-11).

Page 4, line 16: Reword to (?): '…above a coral reef exposed to air…'

We have changed the wording here to '…above a coral reef exposed to air at low tide' (page 4, line 16-17).

Page 4, line 17: Capitalisation of 'Octocorals' necessary? These are animals as in 'dogs'?

Capitalisation of 'Octocorals' has been removed (page 4, line 19).

Page 4, line 18: What are the concentrations for dinoflagellate cells and benthic algae?

Average concentrations for benthic macroalgae and free-living microalgae sampled from the GBR are provided (page 4, line 21-23).

Page 4, line 21: Check structure of information. Consider removing orphan sentence/paragraph.

The orphan sentence has been merged with the following paragraph (page 5, line 16-17).

Page 4, line 24 - 28: This sentence does not make logical sense. It starts off with information on DMSPp but then seems to include info on DMSPd (release from phytoplankton during grazing) and other non-particulate exudates.

$DMSP_w$ refers to dissolved or non-particulate DMSP (rather than $DMSP_d$). Both $DMSP_w$ and $DMSP_p$ may be released from the coral holobiont into reef waters via zooxanthellae expulsion, followed by natural senescence or grazing. This section has been rephrased to make this point clearer (page 5, line 7-12 and Figure 3).

Page 4, line 28 - 29 and page 5, line 4: There is also the possibility that DMS is being produced from DMSP without the production of acrylate (ddd-cleavage pathway): Todd, J. D., Rogers, R., Li, Y. G., Wexler, M., Bond, P. L., Sun, L., Curson, A. R. J., Malin, G., Steinke, M. and Johnston, A. W. B. (2007) Structural and regulatory genes required to make the gas dimethyl sulfide in bacteria. Science, 315, 666-669. (See also P5, L4).

We have described this alternate DMSP cleavage pathway in the text, with reference to Todd et al. (2007) (page 4, line 28-29).

Page 5, line 18: I disagree with: '…Until recently, it was thought that biosynthesis was limited to photosynthetic endosymbionts…' – (i) Many non-endosymbiotic organisms make DMSP and (ii) it has been long known that the heterotrophic dinoflagellate (= 'animal') Crypthecodinium cohnii can make DMSP, for example: Uchida, A., Ooguri, T., Ishida, T., Kitaguchi, H. and Ishida, Y. (1996) Biosynthesis of dimethylsulfoniopropionate in Crypthecodinium cohnii (Dinophyceae). In: Kiene, R. P., Visscher, P. T., Keller, M. D. and Kirst, G. O. (eds.) Biological and Environmental Chemistry of DMSP and Related Sulfonium Compounds. New York: Plenum Press.

This statement refers to the coral holobiont only, for which it was thought that DMS/P biosynthesis was solely attributed to zooxanthellae. This sentence was not meant to imply that photosynthetic endosymbionts are the only source of dimethylated sulfur compounds in the ocean (we have discussed in several places that non-endosymbiotic organisms such as marine macroalgae, free-living microalgae and heterotrophs also produce DMS/P). We have rephrased this sentence to make this point clearer: 'Until recently, it was thought that *biosynthesis in the coral holobiont* was limited to photosynthetic endosymbionts' (page 5, line 27-28).

Page 5, line 27: (and elsewhere in text): Remove italics from 'spp.'

Italics has been removed from 'spp.' throughout the manuscript (page 5, line 31 and elsewhere).

Page 6, line 23: Is this true?: '…These ROS can diffuse from the algal symbiont into coral cytoplasm…' – I wonder whether ROS are too reactive to pass through biomembranes? Isn't this why they are so damaging to biological structures/molecules?

The capacity of reactive oxygen species to diffuse across biological membranes depends on pH. For example, the superoxide anion dominates in pH conditions $> \sim 5$ (as is found in the coral gastrodermis: pH $\sim 7$, Tresguerres et al., 2017) and has a limited ability to diffuse across membranes. However, in the more acidic symbiosome (pH $\sim 4$) hydroperoxyl radicals may diffuse across cell membranes and so, can diffuse from the symbiosome into the coral gastrodermis (page 7, line 6-7)

Page 6, line 32: Is this statement correct? I do not recall that the paper by Hopkins et al. (2016) addresses the effects of SST or salinity.

Hopkins et al. (2016) was referenced here to support the statement that when corals are exposed to a high level of stress, DMS/P concentrations decline as the rate of oxidation to DMSO increases. Hopkins et al. (2016) demonstrated this when corals were exposed to air, as occurs during low tide. SST and salinity thresholds were listed as examples of conditions which can induce this antioxidant response. For clarity, we have also included 'exposure to air at low tide' (page 7, line 17).

Page 7, line 14: Add apostrophe '…hinders corals' ability…'

An apostrophe has been added to 'corals' ability' (page 8, line 3).

Page 8, line 4: What are those 'time scales'?

An exact time-scale can't really be provided here, given that there is significant variability in temperature and irradiance tolerance between coral species and zooxanthellae clades. In the previous sentence we state that relatively small SST anomalies can result in coral bleaching if conditions persist for 'several weeks'. We then state that SST anomalies > 2°C can cause coral bleaching in a shorter amount of time (i.e. < several weeks), depending on the magnitude of SST anomalies and duration of exposure (page 8, line 23-26).

Page 8, line 13: Check hyphenation '…temperature-sensitive species…'

Hyphen has been added to 'temperature-sensitive' (page 9, line 5 and 15).

Page 8, line 15: Check hyphenation '…macro- and microalge…'

Hyphen has been added to 'macro- and microalgae' (page 9, line 7).

Page 8, line 21 - 22: So, this suggests that DMSP may not have a role in conferring temperature tolerance? Reconsider the wording.

The point being made here was that temperature-tolerant zooxanthellae clades have been found to contain less DMSP compared to temperature-sensitive clades, when exposed to the same conditions. This is likely due to differences in temperature tolerance (i.e. temperatures which induce oxidative stress in Clade C endosymbionts, may not stress Clade D, negating the need to produce as much DMS/P). We have reworded this sentence to make this point clearer (page 9, line 14-18).

Page 8, line 25: Change singular/plural: '…However, tolerance thresholds within Symbiodinium clades are highly variable (Klueter et al., 2017) and do not always predict DMSP biosynthesis (Steinke et al. 2011)…'

This sentence has been corrected (page 9, line 16-18).

Page 8, line 26: Steinke et al. (2011) missing from reference list – carefully check all references.

We apologize for this oversight and have carefully checked that all references have been included in the reference list.

Page 9, line 2: Check hyphenation: '…algal-dominated coral-reef communities…. Also, this sentence may require a reference.

Hyphen has been added to 'algal-dominated' (page 9, line 28). There is no need to hyphenate 'coral reef'. This sentence is a hypothetical and builds on the findings of Lin et al. (2016) who found that algae blooms can promote heat absorption at the surface (page 9, line 26-29).

Page 9, line 4: Check hyphenation: '…pH-sensitive coral-calcification rates…'

Hyphenated the words 'pH-sensitive' (page 9, line 31). There is no need to hyphenate 'coral calcification'.

Page 9, line 6: Change singular/plural '…absorbed by the oceans and affect seawater…'

This sentence has been corrected to 'absorbed by the oceans and affect seawater chemistry' (page 9, line 33).

Page 9, line 7: Avoid repetitive word usage '…Increased CO2 levels increase…'

Repetition of 'increase(d)' has been removed: 'Increased $CO_2$ levels *enhance…*' (page 10, line 1).

Page 9, line 14: It may be important to point out that this is for warm-water corals? I suspect that the damage to cold-water corals is much higher still since the dissolution of CO2 into water is enhanced/temperature dependent?

Clarified that we are referring to projected impacts of ocean acidification in tropical coral reefs (page 10, line 8).

Equation 2: Should the reaction be presented the other way around, because the text refers to erosion of calcareous structures?

We have reversed Eq. 2 to illustrate the process of $CaCO_3$ erosion from left to right (page 10, line 12).

Page 9, line 23: Could add effects of temperature/CO2 on DMS/P in isolation and combined? See: Arnold, H. E., Kerrison, P. and Steinke, M. (2013) Interacting effects of ocean acidification and warming on growth and DMS-production in the haptophyte coccolithophore Emiliania huxleyi. Global Change Biology, 19, 1007-1016.

We have discussed the effects of temperature and ocean acidification on DMS/P production in cultured zooxanthellae, highlighting the stronger influence of temperature, with reference to Arnold et al. (2013) (page 10, line 25-27).

Page 9, line 30: May mention that RCP8.5 is an IPCC scenario?

We have included a brief definition of RCP scenarios where mentioned throughout the text (page 10, line 23-24 and page 20, line 8).

Page 10, line 24: Check hyphenation: '…biologically derived negative feedback…'

Hyphenation of 'biologically-derived' has been removed (page 11, line 20).

Page 11, line 3: Change wording '…undersampled ocean regions…'

The phrase 'under sampled' has been changed to 'undersampled' (page 11, line 29).

Page 11, line 8: It is unclear what is meant by '…all of which reduce the thermal capacity of the sea surface…'. How can the surface concentration of DMS, tidal height, etc. reduce the thermal capacity?

This oversight has been corrected and now reads: 'Higher sea surface DMS concentration, SST and wind speed (which increase diffusivity), and reduced tide height, will enhance the sea-air flux of DMS in coral reef waters' (page 12, line 3-4).

Page 11, line 9: Change wording to 'increased diffusivity'?

The phrase 'reduced diffusivity resistance' has been changed to 'increase diffusivity' (page 12, line 3).

Page 11, line 11: I find it difficult to see how this paragraph is relevant to the specific topic/title of manuscript – unless cold-water corals are considered here? The manuscript sometimes lacks focus.

This paragraph discusses predicted changes to DMS sea-air flux from other ocean regions in a changing climate, for which the impacts in coral reefs are largely unknown. Given that the review discusses the link between DMS and atmospheric properties, we think that it is important to mention these predicted changes, to provide context on the impacts of climate change on DMS emissions and how this may influence local or regional climate (page 12, line 6-22).

Page 13, line 32: More info on '…and other volatile organic compounds (VOCs)…' would be useful here. What other gases are being released?

Other VOCs such as isoprene may also be contributing to the seasonality observed in AOD (page 15, line 6).

Page 14, line 5: What is meant by '…tidal lunar cycle…' – is this not the same? Lunar cycle drives the tides?

This oversight has been corrected. Use of 'tidal lunar cycle' has been replaced with 'tidal cycle' (page 15, line 12).

Page 14, line 26: Avoid use of 'extreme' (it's all relative). May be 'high solar irradiance'?

Use of 'extreme' has been replaced with 'high' to describe solar irradiance (page 15, line 33).

Page 15, line 8 - 9: Check hyphenation '…rainfall-sensitive agriculture…'

Hyphen has been added to the phrase 'rainfall-sensitive' (page 16, line 14-15).

Page 15, line 18: '…or condenses onto existing particles…'

'condense' has been changes to 'condenses' (page 16, line 24).

Page 16, line 9 - 10: Coral bleaching is not a stressor '…impacts of environmental stressors such as coral bleaching…'. Rewording necessary.

Coral bleaching was mentioned here as an example of the *impacts* of environmental stressors, not as an example of a stressor. Section 7 has been rewritten (see general comment responses) and this sentence is no longer included (page 17-19).

Page 16, line 10: Check spacing: '… coral reefs is also…'

Section 7 has been rewritten.

Page 16, line 19: Avoid 'extreme' and check hyphenation '…are cost- and resource intensive…'

Replaced 'extremely' with 'very' or 'highly' to describe costs and time associated with field campaigns and have hyphenated the phrase 'cost- and resource intensive…' (page 18, line 13 and page 20, line 26).

Page 16, line 21: What is meant by this '…Additionally, field surveys cannot capture processes that may be occurring down-wind of the substance's origin…'? Numerous atmospheric/marine chemistry studies do exactly this? And how is that true for studies on DMSP? Unclear – would need a re-write or more information?

This section has been rewritten (page 18, line 17-21).

Page 16, line 26: Check hyphenation '…cost- and time efficient.'

Again, we have hyphenated the phrase 'cost- and resource intensive…' (page 18, line 13 and page 20, line 26).

Page 16, line 31: Consider capitalisation '…Hotspot and Degree Heating Week (DHW)…'

Degree Heating Week and Hotspot have been capitalised (page 19, line 5).

Page 16, line 33: Check punctuation '…maximum, which, when accumulated over a 12-week moving window, provide a measure…'

This sentence has been removed from the revised manuscript.

Page 17, line 1: Here and elsewhere, consider presentation of water temperature/degree symbol spacing. It should be 8 ºC but some journals also use 8ºC (never 8º C).

The inclusion of a space between the degree symbol and 'C' was a formatting error which occurred when inserting a symbol in MS Word. We had fixed some of these errors prior to submitting the manuscript. We thank you for pointing out the remaining errors which were not picked up in our revisions (page 6, line 5 and elsewhere).

Page 17, line 20: Check hyphenation '…temperature-tolerant coral species…'

Hyphenated 'temperature-tolerant' (page 19, line 26 and 30)

Section 8 and elsewhere: Check use of tenses. For example '…In a modelled scenario, injecting 5 Tg SO2 annually into the stratosphere above Caribbean coral reefs reduces SST, irradiance and sea-level rise and results in a substantial decline in the number of mass coral bleaching events predicted to occur over the next 50 years (Zhang et al., 2017)…'; '…Similarly, Kwiatkowski et al. (2015) reported that enhancing SO2 concentrations in the tropical stratosphere reduces SST and the risk of coral bleaching over the next 30 years under an RCP4.5 scenario…'; …Latham et al. (2013) found that an enhanced source of sea-spray aerosol over the GBR, Caribbean and French Polynesia offsets the warming effects…' Also: consider use of 'reduce' (could be confused with chemical reduction) and use 'decrease' instead?

Past tense has been consistently used here and elsewhere when discussing results from past publications (section 8, page 19-21).

Page 18, line 1 and 4: Avoid repetitive word usage 'another'

Removed repetition of 'another' (page 20, line 16-18).

Page 18, line 5: Consider rewording '…over reefs of high economic and environmental value.'

Reworded this sentence to '…over coral reefs with high economic and environmental value' (page 20, line 20).

Page 18, line 21: '…There is substantial evidence that coral reefs are strong sources of dimethylated sulfur compounds…' It is true that corals are producing a lot of DMSP. However, our recent attempt to simulate the DMS sea-to-air flux from coral reefs using the 'model cnidarian' Aiptasia (which likely has its limitations…) finds that the flux normalised to sea surface area is lower in coral reefs than the average global oceanic flux: Franchini, F. and Steinke, M. (2017) Quantification of dimethyl sulfide (DMS) production in the sea anemone Aiptasia sp. to simulate the sea-to-air flux from coral reefs. Biogeosciences, 14, 5765-5774.

DMS emissions from the GBR are typically in the range expected for other ocean regions (~100-200 ppt). However, several studies have also found that *Acropora* spp. are very strong producers of DMS/P, with concentrations > 3500 nmol cm$^{-2}$ coral surface (Broadbent et al., 2002). The concentrations reported for *Aiptasia* in Franchini and Steinke (2017) were relatively low. It would be very interesting to investigate the contribution of coral reefs to global DMS sea-air flux estimates using concentrations reported from other cnidarians, particularly *Acropora* corals.

*Acropora* are abundant throughout the Indo-Pacific and likely dominate the coral source of DMS/P in this region (Swan et al., 2017b). Concentrations of atmospheric DMS above the Heron Island coral reef flat reached ~45 nmol m$^{-3}$ (> 1000 ppt) when the corals were exposed to air at low tide (Swan et al., 2017a). This suggests that the sea-air flux of DMS from corals can be a much higher, albeit intermittent, source of DMS than the oceanic signal which is dominated by algal emissions. Other studies estimate that the GBR emits 0.02 - 0.08 Tg S yr$^{-1}$ (Hopkins et al., 2016; Jones et al., 2018). Although this represents < 1% of global DMS sea-air flux estimates, it is still a significant amount of sulfur released from a relatively small region of the ocean, with the potential to influence local or possibly regional climate (page 13, line 2-4).

Page 18, line 31: '…This biogenic aerosol source is in danger of becoming weaker with ongoing coral reef degradation…' Earlier, the case was made for DMS being the same/increasing when seaweeds replace corals?

The point being made here was that ongoing coral reef degradation might lead to the loss of a coral-only source of MBA (i.e. not including other potential sources such as marine algae).

We can see that this point was not clear and so have removed this sentence to avoid confusion (page 21, line 14).

References: Check for use of italics for scientific names (e.g. *Symbiodinium* in Deschaseaux et al. 2014b, Klueter et al. 2017).

Scientific names have been italicised in the reference list where required.

[revised manuscript text omitted]